# Successful captive breeding of a Malayan pangolin population to the third filial generation

Dingyu Yan [1✉], Xiangyan Zeng[1], Miaomiao Jia[1], Xiaobing Guo[1], Siwei Deng[2], Li Tao[3], Xiaolu Huang[1], Baocai Li[1], Chang Huang[2], Tengcheng Que[4], Kaixiang Li[1], Wenhui Liang[1], Yao Zhao[1], Xingxing Liang[1], Yating Zhong[1], Sara Platto[5] & Siew Woh Choo [2,6✉]

Pangolins are threatened placental mammals distributed in Africa and Asia. Many efforts have been undertaken in the last century to maintain pangolins in captivity, but only a few of them succeeded in maintaining and keeping this species in a controlled environment. This study reports the first systematic breeding of the *Critically Endangered* Malayan pangolin (*Manis javanica*) in captivity. Our captive breeding approach successfully improved the reproductive rate for both wild and captive-born female pangolins. From 2016 to 2020, we had 33 wild pangolins and produced 49 captive-born offspring spanning three filial generations. The female offspring further bred 18 offspring, of which 14 (78%) were conceived during the first time of cohabitation with males, and four offspring were conceived during the second cohabitation event, suggesting that they may practice copulation-induced ovulation. We observed that captive-born female pangolins could reach sexual maturity at 7–9 months (n = 4), and male pangolins could mate and successfully fertilise females at nine months age (n = 1). We also observed a female pangolin conceiving on the eighth day after parturition (the fifth day after the death of its pup). Our captive pangolins had a female-biased sex ratio of 1:0.5 at birth, unlike other known captive-born mammals. Also, captive-born pangolins were generally more viable after successful weaning and had a similar gestation length (~185 days) to wild pangolins. Most importantly, we report the first self-sustaining captive population of Malayan pangolins, and this species has an efficient reproduction strategy. These advances provide more comprehensive information for people to understand pangolins, and have implications for conserving endangered Malayan pangolins and providing scientific guidance to the management of other pangolin species.

---

[1] Guangxi Forestry Research Institute, Nanning 530002, P.R. China. [2] Department of Biology, College of Science and Technology, Wenzhou-Kean University, Wenzhou, Zhejiang, P.R. China. [3] Guangxi Institute of Veterinary Research, Nanning 530001, P.R. China. [4] Guangxi Terrestrial Wildlife Rescue Research and Epidemic Disease Monitoring Centre, Nanning, Guangxi 530003, P.R. China. [5] Department of Biotechnology, College of Life Sciences, Jianghan University, Wuhan, Hubei, P.R. China. [6] Zhejiang Bioinformatics International Science and Technology Cooperation Centre, Wenzhou-Kean University, Wenzhou, Zhejiang, P.R. China. ✉email: Yandy6@126.com; cwoh@wku.edu.cn

Pangolins are the world's only scaly mammals, belonging to the Order Pholidota, Family Manidae, with a total of eight extant species in three genera—four from Asia (*Manis javanica*, *M. pentadactyla*, *M. crassicaudata* and *M. culionensis*) and four from Africa (*Phataginus tricuspis*, *P. tetradactyla*, *Smutsia gigantea* and *S. temminckii*)[1]. Pangolins are the most trafficked group of wild mammals, replacing ivory as the world's most sought-after wildlife[2–4], with international trafficking of pangolins and their derivatives between August 2000 and July 2019 estimated to have involved 895,000 individuals. As a result of this immense illicit trade, agricultural expansion and over-harvesting, wild pangolin populations have declined dramatically in recent years[5].

Malayan pangolins (*M. javanica*), which are mainly distributed in Southeast Asia and parts of China's Yunnan Province[6–8], have become one of the three *Critically Endangered* pangolin species listed on CITES Appendix I since 2017[9,10]. Malayan pangolins typically weigh 4–7 kg and have a total body length of up to 140 cm[11]. The gestation period is estimated to be 176–188 days, or ~6 months ($n = 6$)[12], but some researchers believe that the gestation period could be as low as 168 days[13]. With the wild population of Malayan pangolins drastically declining, captive breeding may become a useful means to protect this species from extinction[14].

Over the past 150 years, more than one hundred attempts have been made to maintain pangolins in captivity worldwide. However, only a few of these pangolins have survived for 12–20 years, with most individuals dying within the first few months[15,16]. Researchers attribute this failure to environmental stressors and diseases such as pneumonia, gastrointestinal disorders and infections, likely due to their weakened immune system[14,17–20]. In addition, isolated studies reported pangolin breeding in captivity, but mainly from a small number of wild-caught adults that failed to sire an F2 generation. One exception is a male Chinese pangolin (*Manis pentadactyla*) from Taipei Zoo which has reportedly lived for more than 20 years and has produced a female offspring, which itself has generated two second-generation captive-bred offspring[12,21,16]. Although there is some progress in breeding pangolins in captivity, there is limited knowledge of female reproductive biology including oestrus cycles, uncertainty on gestation period, age at sexual maturity and weaning age. However, breeding pangolins in captivity remains challenging and as a result, they are one of the most difficult mammals to breed in captivity globally.

The current study reports the first successful captive breeding programme for the *Critically Endangered* Malayan pangolin between 2016 and 2020, which is a big step forward to the captive breeding of endangered Malayan pangolins, providing a possible solution for future reintroduction of this species into the wild.

## Results

**Fertility rate of wild female pangolins in captivity**. To examine the reproductive capability of wild female pangolins in controlled environments, we selected 11 wild females for mating. Of the 11 wild females, ten females had 29 pregnancies, producing 30 offspring, including one instance of twins (Supplementary Data 1 and Fig. 1a–e). Of the 30 offspring, 25 were conceived at our centre and five were conceived in the wild and delivered after entering the centre. Although the mating of male and female pangolins in cages was random, the rate of mating and conception was high. For instance, females (WF6, WF8, WF12 and WF16) and males had 30 instances of cohabitation with successful mating. Of these, ten cohabitations during pregnancy were not counted, 16 cohabitations resulted in pregnancy and four cohabitations did not result in conception, resulting in an overall conception rate of 80% (16/20). Of the 16 pregnancies, 13 were conceived during the first cohabitation. Therefore, the proportion of the pregnancies conceived during the first cohabitation was 81% (13/16) (Supplementary Data 1).

**Reproductive rate of captive-born female pangolins**. Of 12 captive-born female pangolins (ten first-generation and two second-generation) that were used for mating, 11 (91.7%) successfully produced 18 viable offspring between August 2017 and November 2020 (Supplementary Data 2 and Supplementary Videos 2, 3). Our data indicate that 18 pregnancies were successfully conceived within 5 days of cohabitation mating (Supplementary Data 2).

The 11 females who produced offspring had 39 cohabitations with successful mating. Of these, 17 cohabitations were not counted as the females mated during pregnancy, 18 cohabitations resulted in pregnancy and four cohabitations did not result in pregnancy, with a conception rate of 82% (18/22). Of the 18 pregnancies, 14 were conceived during the first cohabitation, and four were conceived during the second cohabitation. Therefore, the proportion of pregnancies conceived during the first cohabitation and mating was 78% (14/18) (Supplementary Data 2).

**Sexual maturity in captive-born female pangolins**. We found that 7 out of 11 captive-born female pangolins mated and conceived within a year after their births. One second-generation female pangolin (SG4) was mated and conceived 7 months after her birth, which is the earliest sexual maturity that we observed for second-generation pangolins. Female pangolins FG10, FG15, FG16 and SG4 were first conceived when they were 7–9 months old, even before being weaned from their mothers (Supplementary Table 4). The other seven captive-born females were not caged in time for mating with male pangolins. Therefore, the age of the first conception occurred slightly later between 11–18 months. Thus, our data suggest that captive-born female pangolins could reach sexual maturity as early as 7 months of age (Supplementary Data 2).

**Female pangolins mate with males during pregnancy**. Of 18 pregnant captive-born female pangolins that we observed, half of them mated with males during pregnancy. For instance, the pangolin FG6 produced an offspring on 6 October 2018 after mating four times (in five cohabitations; Supplementary Data 2). Among these four matings, one of them (9–11 April 2018) led to conception, whereas the remaining three matings occurred during pregnancy. The last mating occurred 33 days before parturition. Another example is FG4, which mated with WM8 seven times, 26–29 days before parturition (6–9 March 2018; Supplementary Data 2). Therefore, we estimate that these two females accepted mating 32–53 days before delivery. This mating phenomenon during pregnancy also exists in confiscated wild female pangolins in our centre (Supplementary Data 1).

**Females can conceive soon after the death of their pups**. Pangolin FG6 gave birth on 15 February 2018, and her pup died on 19 March 2018. She mated three times between 9–11 April 2018 and gave birth to a second offspring on 6 October 2018, which indicates that she conceived again 53 days after giving birth to her first pup (22 days after the death of her pup). Similarly, female FG10 mated and conceived 11 days after parturition (8 days after the death of her pup), and female FG16 mated and conceived 8 days after pupping (5 days after the death of her pup). Taken together, we observed that female pangolins can conceive shortly after giving birth or the death of their pups.

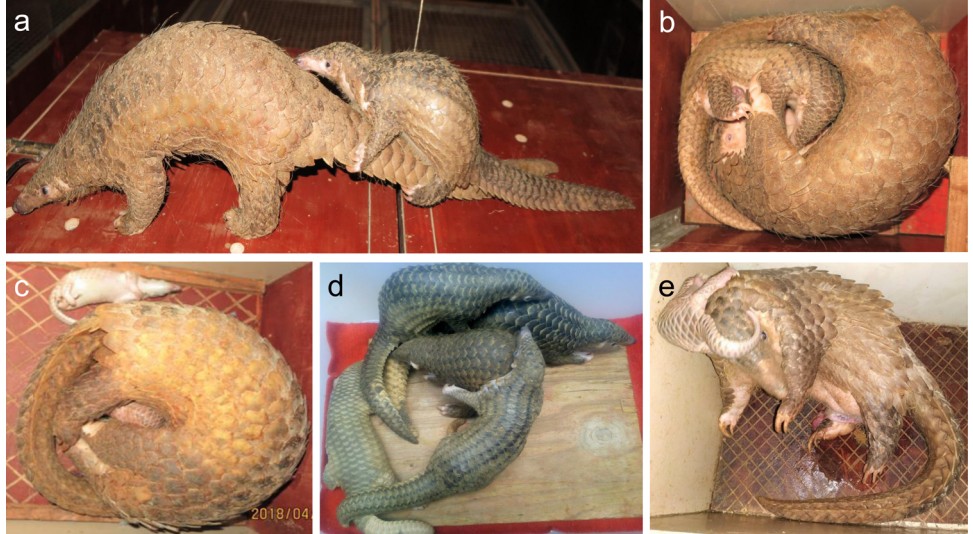

**Fig. 1 Captive-born pangolins. a** Female pangolin WF11 carrying her offspring FG3 on her tail. For more details on how a pangolin carries her offspring, please watch Supplementary Video 1. **b** Female pangolin WF16 nursing her offspring FG5. **c** Female pangolin WF8 was giving birth to a twin FG15.1 and FG15.2. FG15.2 died during birth. **d** A group of the first filial generation pangolins (FG1, FG2, FG3, FG4 and FG5) playing together. **e** Second filial generation female pangolin SG4 after recently having given birth to a third filial generation offspring TG2. The placenta can still be seen attached to the body of SG4. (Photos: Dingyu Yan).

**The proportion of males mating during cohabitation.** To examine the mating willingness of male pangolins, we selected 14 adult wild male pangolins for mating. During cohabitation, males who want to mate usually display behaviour such as following, touching and climbing onto the female before mating commences. Therefore, we considered a male to lack the desire to mate if he did not show any of these behaviours. Of the 14 wild males tested, five (WM6, WM8, WM9, WM11 and M12) showed full mating repertoires from the onset of cohabitation with females. However, most of the males (9 out of 14) showed no intention to mate, which was the same for the captive-born males (4 out of 5). Taken together, we observed that the proportion of male pangolins which mate during cohabitation is low.

**Sex ratio of the captive-born offspring.** Between 2016 and 2020, 44 offspring were conceived in captivity. Among them, the sex of 38 offspring was determined, while the sex of the remaining individuals could not be confirmed as they died in vivo before being fully developed, among other reasons. The sex ratio of the 38 offspring was 24:14 females to males (~2:1 female-to-male ratio).

**Survival rates of wild-born and captive-born offspring.** For the 33 wild pangolins kept at our rescue centre, the annual survival rate improved from 81.8% in 2016 to 100% in 2020 (Fig. 2), with almost half of them ($n = 16$) currently still alive (Supplementary Table 1). We have successfully kept four pangolins for >2000 days, 11 pangolins >1500 days and four pangolins >1000 days.

For captive-born pangolins, the annual survival rates were 85.7% (2016), 60% (2018) and 76.9% (2020). The high survival rate in 2016 might be due to the small sample size ($n = 7$). The rate dropped after 2016 but slightly improved in 2020 after implementing strict measures by the end of 2019. By combining wild and captive-born pangolins, the survival rate was 80% in 2016, increasing to 85.7% in 2020.

Altogether, we have successfully bred 49 Malayan pangolins spanning three filial generations in the 5 years, 20 of which were still alive at the end of this study (Supplementary Table 2).

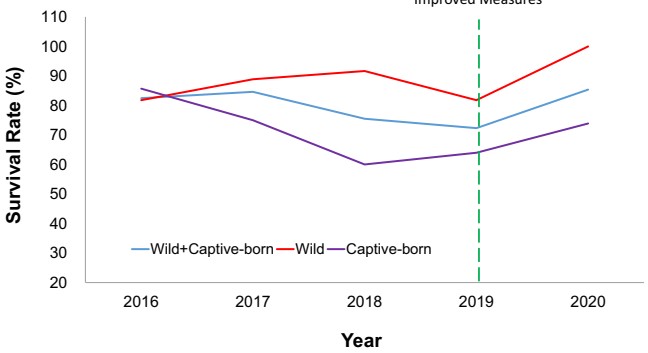

**Fig. 2 Annual survival rate of wild and captive-born Malayan pangolins (M. javanica) from 2016 to 2020.** The green dotted line indicates the time when we implemented strict measures to improve the survival rates of pangolins in captivity.

We kept two pangolins for over 1500 days, eight pangolins for over 900 days, eleven pangolins for over 500 days and eight pangolins for over 150 days. Therefore, we conclude that we can maintain both wild-sourced and captive-bred individuals in captivity with a relatively low mortality rate.

**Survival rate of captive-born pangolins after weaning.** Despite the successes in breeding Malayan pangolins in a controlled environment, captive-born pangolins still have lower survival rates than wild pangolins held in captivity. To investigate the differences in survival rates, we analysed the data from dead pangolins by age group. Published literature reported that the weaning period in Malayan pangolins lasts between 90–120 days[20]. In the current study, however, the weaning period of the captive pangolins was longer, reaching 150 days (manuscript in preparation). Of the 29 captive-born individuals that died, 19 (65.5%) died before birth or pre-weaning (Supplementary Table 2 and Fig. 3a). Encouragingly, the proportion of pangolins that died post-weaning is generally lower than pre-weaning (Fig. 3). Taken together, our results suggest that

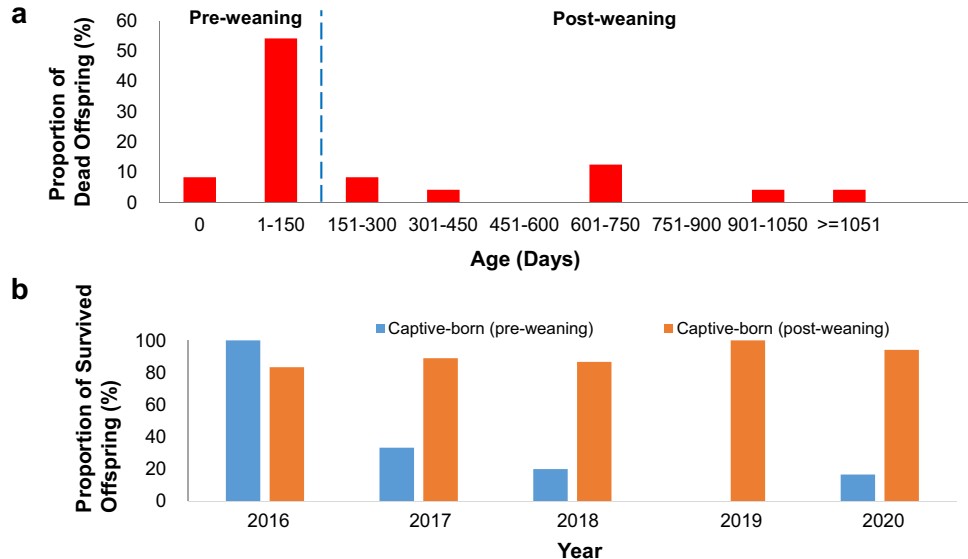

**Fig. 3 Mortality of captive-born Malayan pangolins (*M. javanica*). a** Mortality of captive-born offspring as a function of ages. **b** Comparison between the proportion of surviving offspring between pre-weaning and post-weaning offspring across different years.

captive-born pangolins have a higher survival rate at post-weaning than pre-weaning.

**Assessment of possible cause of death**. To investigate the cause of death in our pangolins, we performed autopsies on 11 adult Malayan pangolins. Necropsy revealed lung lesions (hepatisation) in all pangolins that we examined, as well as two cases of ulcerated foci in the stomach (Supplementary Fig. 1). Bacterial 16 S amplification and sequence analyses revealed the presence of pathogens such as *Morgellons*, *Escherichia coli*, *Klebsiella pneumoniae*, and *Staphylococcus aureus* in the tissues of nine of the pangolins that we examined (manuscript in preparation). Two pangolins that died in our centre in August 2015[22] showed pulmonary bilateral extensive haemorrhagic alveolitis that tended to a red hepatisation, and the molecular results revealed the presence of the pathogen *Morgenia morganii* in both individuals. Therefore, it suggests that these pangolins may have died from bacterial infections.

**Duration of gestation in Malayan pangolins**. To study the duration of the gestation of Malayan pangolins, we calculated the gestation length for each pregnant female pangolin. When females and males cohabited, they often mated several times, and it is, therefore, difficult to determine which mating was successful and led to conception. Therefore, a complete duration of gestation was defined as the interval between the period from the first observed mating to parturition and the period from the last observed mating to parturition. Our data showed that 22 offspring were conceived by mating with wild female pangolins with a gestation range of 154–203 days ($n = 22$; Fig. 4 and Supplementary Data 1). For captive-born pangolins, mating was completed within 5 days and the gestation period ranged from 177–192 days ($n = 17$). By combining the data from all wild and captive-born pangolins ($n = 39$), the gestation period ranged from 154–203 days with a peak at 185 days. The gestation period interval of wild pangolins is broad because we initially did not restrict the interval between the first and last mating during caging, as we did not know how long they had to cohabitate for in order to successfully mate and conceive. This might affect the resolution of gestation length. After further observations, we found that they usually successfully mate and conceive within a

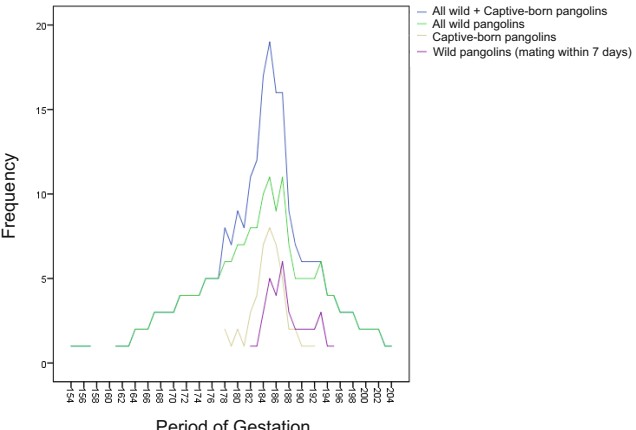

**Fig. 4 Analysis of gestation length in pangolins.** Comparison of the gestation length for different groups of female Malayan pangolins (*M. javanica*).

week. To provide a better resolution of the gestation period interval and to make it comparable to the gestation period of captive-born females, we removed any record of mating exceeding 7 days, resulting in a narrow gestation period of 182–195 days ($n = 10$), which is similar to the distribution of the captive-born females. By combining the data from both groups ($n = 27$), the gestation period ranged from 177–195 days, with a peak at 185 days.

## Discussion

The current study presents a systematic observation of the survival and breeding of Malayan pangolins in captivity. Although previous studies have reported the breeding of pangolins up to the first filial generation (Zhang et al. 2015, 2017), ours is the first study to report captive breeding to the third filial generation and also reports on the highest number of captive-born litters to date ($n = 43$). Our results represent a big step forward in the breeding of pangolins for conservation purposes. Firstly, many research organisations have attempted to maintain wild pangolins in captivity over the past century, but with little success because

captive pangolins usually died of disease. We have managed to keep the small-scale rescued wild pangolins in captivity for an extended period (up to 7 years). Secondly, we have bred captive-bred pangolins up to the third filial generation. Female pangolins generally showed good fertility and reproduced viable offspring. Thirdly, we have established the first self-sustaining captive pangolin population. These advances have implications for the conservation of the Critically Endangered Malayan pangolin.

Since pangolins have a weakened immune system likely due to the loss of interferon epsilon (IFNE) function[2], we recommend that careful attention be paid to potential sources of infections during the rescue and management of captive pangolins, such as the hygiene of the living environment and food. Since December 2019, we took strict measures to ensure that the food and controlled environment were clean by increasing the frequency with which cages were cleaned to 1–2 times per month and disinfecting the shelters with an LPG flamethrower. Moreover, we immediately cleaned the cages when pangolins refused to eat for a day. We replaced nest matting regularly and avoided using medication indiscriminately, including reducing the frequency of antibiotic injections. We used the combination of Amikacin and Kanamycino antibiotics (0.1 mL/kg for each antibiotic) immediately if a pangolin had no appetite to eat for 3 days. We also improved the food storage conditions and placed desiccants in feed cabinets to prevent the growth of pathogens. Encouragingly, we observed no deaths for adult wild pangolins in 2020 (Fig. 2). The annual survival rate of wild pangolins increased from 81.8% (2019) to 100% (2020), and the annual survival rate of captive-born pangolins increased from 64% (2019) to 73.9% (2020) (Fig. 2).

In addition to infections, we believe that successful weaning is also crucial to the survival of pangolins. The high mortality rate of captive-born pangolins during the pre-weaning period might be associated with environmental, nutritional or genetic factors, or the nursery environment. Further research on these factors/issues are required. Also, the mother's stressors may be important during weaning. We, therefore, suggest providing extra care for the mother and baby, which may also improve the survival rate of pups.

On the other hand, pangolins may have endogenous evolutionary mechanisms to maintain their population. Our study revealed that female pangolins can reach sexual maturity at 7 months of age. Zhang et al. (2015, 2017) also inferred that the sexual maturity of Malayan pangolins occurred at ~1 year, even it could occur as early as 6–7 months, based on two instances of wild individuals with low body masses being pregnant[12,21]. Our data showed that captive Malayan pangolin reproduction is aseasonal and that pangolins have no obvious oestrus feature, oestrum or sexual cycle. Therefore, early sexual maturity is not a seasonal phenomenon.

Under our controlled environment, both wild and captive-born female pangolins had a considerably high and efficient reproduction rate. One of the unexpected observations is that female pangolins could mate and conceive in a short period (within 5 days) after cohabitating with males. For instance, 14 of the 18 pups of the captive-born females were conceived during the first random mating with males or at any time when they mate. Therefore, we suggest that pangolins may practice copulation-induced ovulation like European rabbits (Oryctolagus cuniculus)[23,24], which could be a strategy to enhance their reproductive rate. We also observed that female pangolins could conceive shortly after giving birth or the death of their pups. Although the detailed mechanism is unknown, we believe that it may be another reproductive strategy to maintain and expand the species' population. This suggests that this species has an efficient strategy of reproduction.

Our study showed that some male pangolins have a low mating willingness, which may be one of the reasons for the difficulty in breeding pangolins in captivity. The exact reasons for this low mating willingness of captive pangolins and whether wild male pangolins also have reduced mating willingness remain unknown. However, we believe that the low willingness to mate might be related to environmental and/or genetic factors, although more research is needed in the future. Notably, like captive pangolins, captive giant pandas (Ailuropoda melanoleuca) also have a low proportion of males willing to mate. From 1980 to 2019, there were nearly 600 captive giant pandas worldwide, but only 26 captive males could produce offspring through natural mating[25]. It is believed that the low reproduction success rate of captive giant pandas is related to the loss of their natural mating ability, the lack of complete courtship competition and poor sexual desire caused by the process of mate selection especially a breeding strategy may force giant pandas to mate with disliked partners.

Many captive species such as Jaguarundi (Herpailurus yagouaroundi), red panda (Ailurus fulgens), lion (Panthera leo), blackbuck (Antilope cervicapra), leopard (Panthera pardus orientails) and southern pudu (Pudu puda) show male-biased sex ratios, but female-biased ratios are rare[26]. For instance, of the 27 young jaguarondis born in Rotterdam Zoo, only six (22%) were females. Faust and Thompson (2000) analysed the sex ratios of 66 captive mammalian species and found only two species that showed female-biased sex ratios[27]. One of these is the pygmy hippopotamus (Choeropsis liberiensis), which has a significant female-biased sex ratio at birth (59% females), exceeding many other known distorted sex ratios in captive mammals[28]. The common distorted male-biased ratios in captive animals could be a challenge to captive breeding management since it may limit the population growth. Remarkably, our data showed that the pangolin population had a highly female-biased sex ratio (63% females) at birth, probably due to the influence of environmental factors and/or the artificial food that we fed. This ratio has implications for captive management and is advantageous since our methods produce more females than males, which may eventually lead to the rapid increase of their population. Notably, we cannot rule out the possibility that this result may still be preliminary because of its small sample size. It would be interesting to calculate the sex ratio using a larger sample size to confirm this observation.

The main objective of this study is to prove that maintaining and breeding pangolins in captivity is feasible. Although we have established a captive pangolin population, there is still room for improvement. Firstly, it would be interesting to evaluate the genetic structure of these captive pangolins and compare this with wild pangolins. Secondly, to avoid potential inbreeding and further enhance our breeding programme, we are planning to source additional males and carry out related research to determine the reasons for some males' unwillingness to mate. Thirdly, a key focus of future research should be on further understanding the conditions necessary for survival and successful breeding in captivity. We are also improving the management of pre-weaning pangolins and nutrition given to captive pangolins to further enhance our breeding programme. Fourth, we will take captive pangolins to a designated wild area (6.6 hectares of coniferous and broad-leaved mixed forest) to expose them to more natural conditions to acquire the ability or skills to survive. After strict observation and evaluation, we will release them into the wild according to the IUCN Guidelines for Reintroductions, including subjecting them to a comprehensive disease screening and genetic diversity analysis. One of the notable reintroduction success stories is that of European bison (Bison bonasus). European bison were hunted to extinction in the wild and survived only in captivity in the early 20th century, but were successfully reintroduced

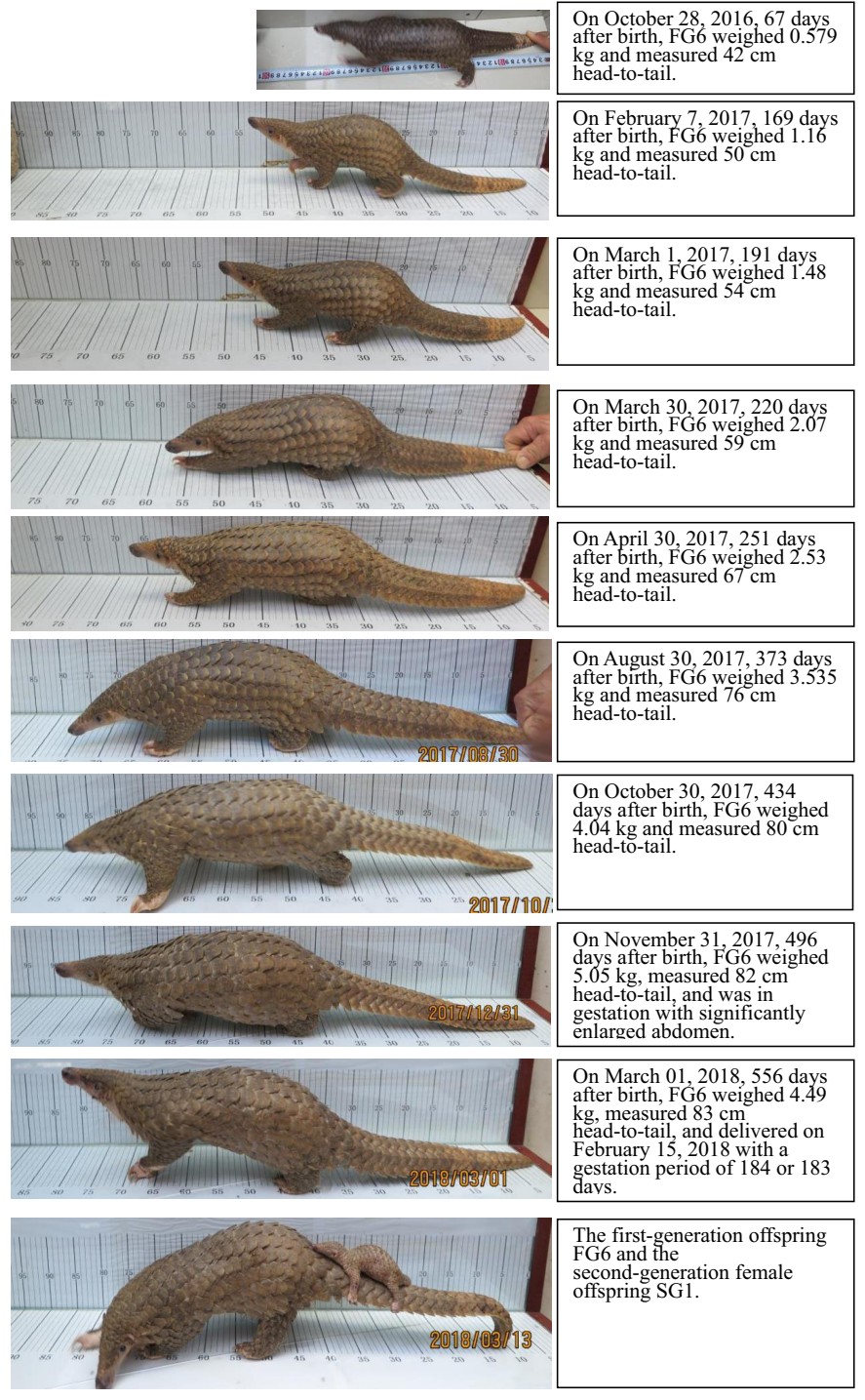

**Fig. 5 Growth rate in pangolin.** Growth process of the first-generation Malayan pangolin (*M. javanica*) offspring FG6. (Photos: Dingyu Yan).

to the wild in the 1950s[29,30]. Therefore, we believe that our captive pangolins can serve as genetic reserves for re-establishing natural populations and ameliorating the conditions of wild pangolins.

In conclusion, our study sheds new light on the captive breeding of endangered pangolins and provides a reference for the management of other captive breeding initiatives for the conservation of endangered species.

## Methods

**Animal ethics.** This project was approved by the Forestry Department of Guangxi Zhuang Autonomous Region, Guangxi Province (domestication and breeding

permit of national key protected wild animals; Permit Number: A2016008). All works and protocols were approved by the Biology Ethics Committee of the Guangxi Forestry Research Institute [Reference Number: GXFI (A2016006)].

**Animals used in this study.** The 33 Malayan pangolins used in this study were of wild origin and were confiscated by Chinese law enforcement officers (Supplementary Table 1). Each female was given a code represented by two letters and a progressive number (WF1, WF2, etc.). The detailed information on the mating and offspring are presented in Supplementary Tables 2 and 3. To better manage and control this experiment, we stopped receiving any wild pangolins into our centre since the beginning of this study in 2016.

**Animal housing, husbandry and care.** Pangolins were kept in indoor cages, each consisting of three areas: activity area (120 cm × 80 cm × 50 cm), insulated wooden

winter den (40 cm × 35 cm × 28 cm), and underground summer den (40 cm × 35 cm × 28 cm). Each pangolin was kept in a single cage. For the purpose of maintaining hygiene, the activity area was raised by 5 cm and was fitted with a wire mesh floor to enable faeces to fall through. Malayan pangolins originate from tropical regions of Southeast Asia such as Malaysia and Indonesia, therefore to mimic their natural climate, each wooden winter den (including a blind recording area) was equipped with a temperature control device to keep the temperature at 24–26 °C during winter. Pangolins could travel freely between these three areas. This setup allowed pangolins to select which den to inhabit based on seasons and indoor temperatures. To minimise the risk of infections, the wooden plate below each cage, as well as the wire mesh of all cages, were cleaned monthly with clean water using a high-pressure water gun, and the dens were disinfected with a liquefied petroleum gas (LPG) flamethrower (a heating or welding tool that used for burning) while the pangolins were not in occupation.

All pangolins were fed with a special formula consisting of black ant powder, silkworm pupae powder, mealworm powder, soy protein powder, termite mound mud and a small amount of vitamin complex with a ratio of 40:20:30:8:2:0.1. The formula was mixed with water and blended until reaching a semifluid consistency. Pangolins were fed with 250–400 mL of formula once a day between 17:00–20:00. Clean water (100–200 mL) was also provided in a separate bowl daily. All food was stored in a dry and hygienic environment to minimise the growth of pathogens.

At the end of this project, all living pangolins continued to be maintained in our centre for a long-term study. The individuals that are suitable for release will be gradually adjusted to living in an outdoor environment.

**Animal monitoring**. The experiment was conducted using surveillance video playback to record pangolin mating behaviour. As female pangolin's oestrus is not obvious, we randomly (cohabitation time was random) placed two individuals (one male and one female) in the same cage for mating, taking into account the identities and personal information of each pangolin to avoid inbreeding. Two cages were placed against each other, with the two activity areas abutting and a door in between. When individuals were placed together to cohabitate, we opened the door leading between the two communal areas. A monitor was placed in the common area that was shared by both individuals to continuously record their activity. Whether mating occurred successfully was determined by the observed behaviour. Important information such as the cohabitation time, the mating date and time, the number of mating events, the date of parturition and the gestation period were recorded. Every newly-born offspring between 2016 and 2017 was weighed on the 10th, 20th and 30th day of every month, and their total length (measured from the tip of the snout to the tip of the tail with the individual stretched out; Fig. 5) at the end of every month (30th day).

**Gestation length**. When a female and male p were placed together in a cage, they sometimes mated several times, so it was difficult to determine which mating led to conception. Therefore, the gestation length was estimated as the time interval between both the observed first mating to parturition and the observed last mating to parturition. We observed ten instances in which the females cohabited and mated only once, and the gestation period of these instances ranged between 178–193 days (defined as the basic gestation period). For other females who cohabited and mated twice or more before parturition, we calculated the gestation period based on the birth of the pup and the basic gestation period as a reference. For example, a female cohabited and mated twice (on 31 December 2019 and 3 March 2020) before giving birth on 1 July 2020. If the first cohabitation resulted in conception, the gestation period was 183 days, whereas it was 120 days if the second cohabitation resulted in conception. Since the 183 days is within the observed basic gestation period and the 120 days fall well outside of this range, we considered the first cohabitation to have resulted in conception, and the gestation period, therefore, being 183 days.

**Dissection of dead animals and identification of pathogens**. We dissected 11 dead pangolins (out of 44 pangolins that died across the study duration) in Guangxi Institute of Veterinary Research and visually inspected them for any anomalies which may indicate the cause of death. Lung and liver samples were excised under hygienic conditions and total genomic DNA was extracted following the guidelines of the bacterial genomic DNA extraction kit (CWBIO, China). We used the universal bacterial 16 S primers 27 F (AGAGTTTGATCMTGGCTCAG) and 1492 R (GGCTACCTTGTTACGACTT) to amplify bacterial DNA. PCR reactions were performed in a 50 μL reaction volume, including 3 μL of template DNA, 2 μL of each primer (10 μmol/L), 26 μL of 2×Taq PCR Master Mix (Beijing Cowin Biotech Co., Ltd), and added ddH2O to a total volume of 50 μL. The PCR reaction was performed for 35 cycles, including initial denaturation at 95 °C for 5 min, denaturation at 95 °C for 1 min, annealing at 60 °C for 1 min, extension at 72 °C for 1 min and final extension at 72 °C for 10 min. PCR amplification products were validated using 1.5% agarose gel electrophoresis. The gels were recovered, and the PCR products were sequenced and analysed by Invitrogen Co., Ltd. (Shanghai).

**Reporting Summary**. Further information on research design is available in the Nature Research Reporting Summary linked to this article.

**Data availability**
Data supporting the findings of this study are included in the published paper and supplementary files. For additional information, please contact the corresponding authors.

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

## Acknowledgements

We would like to thank the Department of Wildlife Protection of State Forestry and Grassland Administration, China Wildlife Protection Association, and Guangxi Forestry Bureau for funding and full support throughout this study. We thank the Guangxi Terrestrial Wildlife Rescue Research and Epidemic Disease Monitoring Centre for providing animals for this study. This study was funded by the Nature Science Foundation of Guangxi (2018GXNSFAA294066), the State Forestry Administration of China (Reference number: 2019072), Guangxi Forestry Bureau (Reference numbers: GL2018kt-17 and GL2020kt-25). Huang Chang and Siwei Deng were funded by the high-level talent recruitment programme for academic and research platform construction (Reference number: 5000105) from Wenzhou-Kean University.

## Author contributions

D.Y. conceived this project. D.Y., K.L., X.Z., B.L., W.L. and L.T. performed experiments and collected data. D.Y. and S.W.C. performed data analyses and interpretation. D.Y., S.W.C., S.D., and Y.Zha. wrote the manuscript. M.J., X.G., S.D., X.H., C.H., T.Q., X.L., Y.Zho. and S.P. revised the manuscript.

## Competing interests

The authors declare no competing interests.
