## [Peer Review File · Communications Biology]

Reviewers' comments:

Reviewer #1 (Remarks to the Author):

In this manuscript, Dingyu et al. describe the establishment of a self-sustaining captive pangolin population. This is exciting because pangolins have been generally difficult to maintain in captivity, despite the increasing need to support wild populations. As this manuscript is descriptive in nature, I have no issues with the design of the project. However, I do think a few things require a bit more explanation.

From the many descriptions in the text, these pangolins will readily mate and produce offspring in the captive environment. What was done differently here that made this captive population work compared to all the previously failed attempts? I understand that the strict cleaning and sterilization measures implemented in 2019 increased survival after that time, but what led to the increased survival relative to other programs before 2019? More explanation of the differences between other and this project are needed to really understand what caused this so the techniques could be applied to other populations and species.

It is interesting that even in only 2 generations, survival of captive individuals is less than survival of wild individuals in captivity. Typically, I think of survival in captivity as being equal among captive and wild born individuals or perhaps favoring captive born individuals via adaptation to the captive environment over time. Do you have some explanation (or hypothesis) that can explain why captive born individuals did not survive as well as wild born individuals in captivity?

Finally, I understand that the mating pairs were random because you lacked any existing data on the relatedness between most wild-born individuals. However, it isn't entirely clear how the available information was used. For example, lines 110-111 say that captive born males and females were used for mating; were these full or half siblings? In addition, you can now make some calculations about founder genome equivalents that are present in your current captive population and use these to lay out your plans for managing this population moving forward and provide some hints as to the relationship of the captive born individuals. For example, how many individuals are full or half siblings? Will you initiate a minimizing mean kinship approach for selecting mates in future generations, now that you know parentage? How many males will you need to add to your population to reach your desired population size while maintaining genetic diversity? Adding these details will help show that you have indeed produced a self-sustaining pangolin captive population.

Minor comments:

Lines 19-20: Something seems off grammatically with this sentence.

Line 75: Should "to farming" be removed? If not, this needs more explanation.

Line 84: I think this should be, "Here, for the first time, we report a large-scale..."

Line 84: I do not think that 5 years/3 generations is "long-term" please consider rephrasing this

Line 105: How were confiscated pangolins distributed across China? Were they all from the same locality?

Line 108: Were the 5 litters born in the field considered wild born and were they ever mated together? I know the mating was random but it seems that sibling pairs should be avoided.

Line 145: Later on you explain what this training is but it is confusing here. I suggest you add some language that you will expose pangolins to more natural condition prior to releasing them back into the wild.

Here forward I have not noted line or page numbers because they are absent:

"we bred the offspring and calculated its successful rate" I am not completely sure what you mean here, maybe 'success rate' instead of 'its successful rate'?

"August 2017 to November2020" missing a space between November and 2020

"For instance, FG3 and FG10 bred three offspring, FG6 and FG16 bred two offspring, and the other five captive-born pangolins bred an offspring." I am not familiar with this kind of use of the word 'bred'. In this sentence I would use 'produced' instead.

Reviewer #2 (Remarks to the Author):

This manuscript reports on the breeding success of 11 wild-caught Sunda pangolins (*Manis javanica*) in captivity, together with the breeding success of their captive-born offspring. Although previous studies have investigated captive breeding success, this is the first study that I know of that has done so in a systematic way and has reported successful breeding of the F2 generation.

However I found this manuscript overall to be fairly poorly written, and often difficult to follow. In your Introduction, you need to introduce your study species in more detail, specifically including details such as published average and maximum masses and lengths, and published gestation periods. This will allow readers to better interpret your results.

I found the Methodology to be confusing. For example, I am unsure whether male and female pangolins were housed separately on two ends of a cage with an intervening 'common area', or whether they were placed together in a single cage during copulation. In your results section you also mention that bacterial diversity in the lung tissue of pangolins that had died was investigated using molecular techniques, but this is not mentioned at all in your Methodology.

I also found the Results section difficult to follow, mainly because the results are often analysed and discussed directly after being mentioned.

I think that the quality of this manuscript would be greatly enhanced if the results are separated and treated discretely in the Results and Discussion sections, i.e., use the Results section just to report your Results, and use the Discussion to analyse and compare these results to previous studies.

Please find below a list of specific queries, concerns and comments that I have:

1. Throughout the manuscript, use an en-dash (not a dash) to indicate a range of values.
2. Line 1: Change the title to "Systematic captive breeding of Malayan pangolins: implications for conservation"
3. Line 19: Replace "endangered" with "threatened". "Endangered has a very specific meaning in conservation, as it refers to a category on the IUCN Red List. As the eight pangolin species are variously classified as Vulnerable, Endangered and Critically Endangered on the IUCN Red List, using the word "endangered" in this context infers a false meaning.
4. Lines 19-20: Replace "carried out to 20 captive pangolins since hundred years ago" with "undertaken in the last century to maintain pangolins in captivity".
5. Line 22: Replace "mating reproduction" with "breeding".
6. Lines 25-26: Replace "10 of them successfully produced" with "10 of successfully produced".
7. Line 34: Replace "9-month" with "9-months".
8. Line 35: Replace "captive pangolin" with "captive pangolins"
9. Lines 38-39: Replace "have similar gestation length" with "have a similar gestation length"
10. Line 39: Replace "with the wild pangolins." with "to the wild-caught pangolins."
11. Lines 50-51: Replace "Pangolins are the most endangered wildlife species and recognized as the most 51 ravaged mammal by illegal trade" with "Pangolins are the most trafficked group of wild mammals".
12. Lines 52-53: Replace "There are eight modern pangolin species" with "There are eight extant pangolin species"

13. Lines 53-55. These common names are very outdated. Use the most recent common names as accepted by the IUCN and published on the IUCN Red List, and also include the most recent scientific name for each species after the common name.
14. The in-line referencing needs to be corrected. For example, in some instances the author's surname and year are provided, but in other instances (for example line 56) the authors' full name, abbreviated second name and surname are provided, in addition to the year ("Timothy J. Gaudin 2009").
15. Line 57: Why mention where Chinese pangolins occur? You did not use them in your experiment, nor did you mention where the remaining six species occur. Remove this section.
16. Lines 56-57. Include additional information on the Malayan pangolin, specifically what is the average and maximum mass, and the average and maximum body length? This will help readers to determine whether the individuals that you used in your experiments were juveniles, young adults or mature adults. Also include what is currently known about their gestation and weaning periods.
17. Line 58: This reference is very outdated. The most recent IUCN conservation assessment is from 2019 – use that.
18. Line 63: Merge these references to read: "Choo et al. 2016, 2020; Challender et al. 2019"
19. Line 64: Change "wildlife" to "wildlife species".
20. Line 65: Change "extinction" to "continued existence"
21. Line 74: Additional references that you should include are Hua et al. 2015, and Hoyt, R. 1987. Pangolins: Past, present and future. Proceedings of the AAZPA National Conference: 107-134.
22. Line 75; Replace "farming" with "captive maintenance". None of the references that you cite here specifically looked at the challenges to farming pangolins. Rather, they all considered the challenges to maintaining pangolins in a captive environment.
23. Lines 77-78. Again, in-line references need to be standardised. Here the author's initial is sometimes given, while at other times it is not.
24. Lines 78-81. This sentence needs to be reworded, as in its current state it is very difficult to understand. Also, in line 81, combine these references.
I suggest changing it to: "Although there have been isolated reports of pangolins breeding in captivity, these offspring were born from a small number of wild-caught adults and these offspring failed to sire F2 generation offspring after having been reared in captivity (Zhang et al. 2015, 2017)."
- Lines 82-83: Again, this sentence is unclear and needs to be reworded. I suggest changing it to: "Therefore, breeding pangolins in captivity remains challenging and as a result they are one of the most difficult mammals to breed in captivity globally".
25. Lines 84-85 need to be reworded. I suggest "We report on the first large-scale and long-term captive breeding program for critically endangered Malayan pangolins spanning a 5-year period from 2016 to 2020."
26. Lines 85-93 (starting at "Our data..."). These are Results and Discussions and should not be included in your Introduction. Remove this section.
27. Lines 104-106. Reword to: All 33 Malayan pangolins used in this study were of wild origin and were confiscated by law enforcement officers from smugglers in China (Supplementary Table 1)."
28. Lines 106-111. Replace "litter" with "offspring".
29. Lines 106-111 needs to be reworded. I suggest: "By November 30, 2020, we had 48 offspring born at our rescue centre at Guangxi Academy of Forestry Science, of which 46 were normal births (Supplementary Table 2). Among of these, five offspring were conceived in the wild but were born at our centre. The remaining 43 offspring were conceived at our centre, and of these 41 were normal births, with a single instance of twins being born. Eleven wild females, three wild males, twelve captive-born females and a captive-born second-generation male were used for mating (Supplementary Table 3)."
30. Line 112: Change "large scale and long-term experiments" to "large-scale and long-term experiment".
31. Line 113: Change "base" to "centre".
32. Line 116: Change "are originated" to "originate".
33. Line 117. Capitalize "Indonesia".
34. Line 117: Change "climate" to "climates".
35. Line 118: Add a comma after "zone".
36. Line 119: Change "the indoor cage" to "indoor cages".
37. Line 128: Delete "of pangolins".

38. Line 129: Change "setting" to "settings"
39. Line 138: Change "fodder" to "food".
40. Line 139: Change "fluid" to "liquid".
41. Lines 140-141: Change "Although the food had liquid, clean water was still provided in a separate bowl because some pangolins might need to drink it." to "Although the food was liquid, clean water was still provided ad lib in a separate bowl."
42. Line 142: Change "all food were stored at" to "all food was stored in a"
43. Lines 144-145: Change "At the end of this project, all living pangolins are maintained in our base for a long term observations and study." to "At the end of this project, all living pangolins continued to be maintained in our centre for a long-term observational study."
44. Lines 145-146. What do you mean by "Some of them will be subjected for training before we release them into the wild.?" Do you mean that you will be training some of these individuals, or will they be used to teach/train people? If the latter, what type of teaching will they be used for?
45. Lines 151-153: Change "The doors of cages kept open and monitor was placed towards to the common room that they both stayed in order for continuously recording for 24 hours." to "The cage doors were kept open and a monitor was placed in the common room that they both shared in order to continuously record their activity for 24 hours."
46. Lines 151-153. This sentence needs to be clarified. The way it is described is sounds like there are three cages: one cage on either side housing the two individuals separately, and a central "common room" that both individuals have access to once their respective cage doors are opened. Is this the case? Or were both individuals placed in a single cage?
47. Lines 153-154: Change "Whether the mating was successfully done or not were determined according to their mating actions." to "Whether mating successfully occurred was determined by their mating behaviour."
48. Line 155: Change "the number of mating" to "the number of mating events".
49. Line 156: Change "date of calving and the pregnancy period" to "date of pupping and the gestation period"
50. Line 158: What do you mean by "early-born offspring"?
51. Line 159: Change "measured for their head and tail lengths" to "their total length measured".
52. Line 159: How was their total length measured? Was it measured while they were curled up, or while they were stretched out straight? These two methods result in different measurements, so the method used needs to be stated.
53. Line 173: Change "environment" to "environments".
54. Line 177: Change "the base" to "our centre".
55. Lines 179-180: "our data showed that the rate of mating and conception was generally high". Where are these data? If you have presented them, then you need to include a reference to the table or figure. What do you mean by high? You need to include values here like how many individuals mated in how many experiments, and of those mated how many led to pregnancy? I assume that these data are all captured in Table 1, but you need to summarise them here as well so that the reader does not have to try and infer this information from your Table 1.
56. Lines 182-183: As mentioned above, you cannot make this statement unless you include data on the actual numbers of attempted matings, successful matings and the proportion of individuals successfully conceiving and giving birth.
57. Figure 1a. I do not understand why this figure or the associated video have been included. It is a well-known fact that pangolins carry their offspring on their tails (see for example introductory chapters and species accounts in Challender, D.W., Nash, H.C. and Waterman, C., 2019. Pangolins: Science, Society and Conservation. Academic Press, and references therein).
58. "Captive-born female pangolins have good reproductive success", Line 2: Change "in artificial environment" to "in an artificial environment".
59. "Captive-born female pangolins have good reproductive success", Lines 2-3: Change "calculated its successful rate" to "calculated the success rate".
60. "Captive-born female pangolins have good reproductive success", Lines 3-6: This section is confusing. In lines 3 & 4 you say that there were 12 F1 offspring, and that these produced 18 viable F2 offspring. But in lines 5 & 6 you say that there were 10 F1 offspring, and that they produced 15 F2 offspring. Are lines 5 & 6 supposed to refer to F2 (i.e., second generation) offspring?
61. "Captive-born female pangolins have good reproductive success", Line 5: Change "November2020" to "November 2020".
62. "Captive-born female pangolins have good reproductive success", Lines 6-8: Remove the

sentence starting "For instance,..." and ending "bred an offspring". These details are captured in Table 1, and do not provide any additional information that needs to be repeated in the text.

63. "Captive-born female pangolins have good reproductive success", Line 12: What do you mean by "artificial breeding"? Artificial breeding means the artificial insemination of a female using sperm that was harvested from a male. This has not been mentioned at all in this manuscript, and I therefore assume that "artificial breeding" is used in error here.

64. "Captive-born female pangolins have good reproductive success", Lines 13 & 14: You cannot say that your diet and conditions are "suitable for the normal growth of these captive pangolins" as nowhere have you compared the growth rates of these captive-born pangolins to the growth rates of wild-born pangolins. At best you can say that your conditions are suitable for rearing pangolins.

65. Figure 2: Remove "Photograph" from all your captions. i.e., Caption 1 should just read "On October 28 2016,....", Caption 2 should read "On February 7 2017, ...", etc.

66. Figure 2: Your caption "A characteristic white tail, a few dark scales among white scales" does not add any value as it in no way addresses the topic of your study, i.e. breeding pangolins in captivity. Remove this caption.

67. Figure 2 legend. Change your current legend to "Figure 2. The growth process of the first-generation offspring FG6. Each captive-born pangolin was regularly weighed and measured (Photos by Yan Dingyu).

As mentioned above, your statement "Some unique features of pangolins were also highlighted." adds no value as it has nothing to do with breeding pangolins in captivity. Also as mentioned above, you cannot say that "FG6 grew normally" as you did not compare her growth rates to those of wild pangolins.

68. "The first cage mating has a high conception rate", line 3: Change "produced" to "conceived".

69. "The first cage mating has a high conception rate", line 9: Change "need" to "needs".

70. Table 2: Change "Times of Mating" to "Number of mating instances".

71. "Captive-born female pangolins reach sexual maturity in 7-8 months", Lines 3-5: Replace "For instance, the second-generation SG4 had mated and conceived at 7th month after her birth, which is the earliest sexual maturity that we observed for second-generation pangolins." with "The second-generation SG4 mated and conceived 7 months after her birth, which is the earliest sexual maturity that we observed for second-generation pangolins."

72. "Captive-born female pangolins reach sexual maturity in 7-8 months", Lines 5-7: Replace "For female pangolins FG10, FG15, SG4, and FG16, they were first conceived at 7-9 month old, even before the separation from their mothers." with "Female pangolins FG10, FG15, SG4, and FG16 first conceived when they were 7-9 month old, even before being weaned from their mothers."

73. "Captive-born female pangolins reach sexual maturity in 7-8 months", Lines 6-7: Why were these young pangolins (7-9 months old) still with their mothers? I assume that this is because they were not able to disperse naturally because of their confined conditions? Various authors (e.g., Lim & Ng, 2008; Nguyen et al. 2014) report that weaning occurs at 3-4 months, suggesting that these offspring should have already been independent of their mothers. [Lim, N.T. and Ng, P.K., 2008. Home range, activity cycle and natal den usage of a female Sunda pangolin *Manis javanica* (Mammalia: Pholidota) in Singapore. *Endangered Species Research*, 4(1-2), pp.233-240. and Nguyen, V.T., Clark, V.L., Tran, Q.P., 2014. Sunda Pangolin (*Manis javanica*) Husbandry Guidelines. Carnivore and Pangolin Conservation Program, Save Vietnam's Wildlife, Vietnam.

74. "Captive-born female pangolins reach sexual maturity in 7-8 months", Line 9: Change "the age of first conception was occurred slightly late between 11 to 18 months" to "the age of first conception occurred slightly later between 11-18 months".

75. "Captive-born female pangolins reach sexual maturity in 7-8 months", Lines 11-13: Change "as earliest as approximately 7-8 months after their births. Zhang, et al. (2015) inferred that sexual maturity of the Malayan pangolin occurred at approximately 1 year, even it could occur as early as 6-7 months" to "as early as 7 months of age. Zhang et al. (2015) inferred that sexual maturity of the Malayan pangolin occurred at approximately 1 year, although also reporting an instance of a captive female being pregnant at 6-7 months of age"

76. "Female pangolins accept mating with male pangolins during pregnancy", Lines 3-5: These data do not match what is in Table 1. For instance, Table 1 says that she was born on 22 August 2016 (not 6 October 2018),

77. "Female pangolins accept mating with male pangolins during pregnancy", Line 3: Change "For instance, the pangolin FG6 produced on 6 October, 2018 and mated with male four times (in five times of caging) before calving. Among of these four mating, one of them (9-11 April 2018) led to conception, whereas the reaming three mating occurred during pregnancy. The last mating

occurred 33 days before calving with the male pangolin WM9. Another example was FG4 mated with WM8 seven times in 26-29 days before calving (from 6 to 9 March 2018) (Table 2). Therefore, we estimated that the two females accepted to mate 53–32 days before calving. This phenomenon of mating during pregnancy also exists in wild female pangolins (Table 1).” to “For instance, pangolin FG6 pupped on 6 October 2018 after having mated four times (during five caging events). Among these four matings, one of them (9–11 April 2018) led to conception, whereas the remaining three matings occurred during pregnancy. The last mating occurred 33 days before pupping. Another example was FG4, which mated with WM8 seven times in 26–29 days before pupping (between 6 and 9 March 2018; Table 2). Therefore, we estimate that the two females accepted matings 53–32 days before pupping. This phenomenon of mating during pregnancy also exists in wild female pangolins (Table 1).”

78. “Female pangolins accept mating with male pangolins during pregnancy”, Line 6: Change “reaming” to “remaining”.

79. “Female pangolins accept mating with male pangolins during pregnancy”, Lines 10-11: “This phenomenon of mating during pregnancy also exists in wild female pangolins”. What is the reference for this statement? If you say that females in the wild also mate while they are pregnant, you need to provide evidence or a reference to support this.

80. Table 1, 2 and throughout the manuscript: If females were placed with males multiple times before parturition, how do you know that they always conceived during the first pairing? Why could it not have been during the second or subsequent pairings? If you were using gestation period to determine likely date of conception, then you cannot report on gestation period here as that then becomes a circular argument. How do you know that gestation period isn’t actually a lot shorter than what you record, suggesting that females only became impregnated during a subsequent mating event?

81. “Female pangolins conceive within a short period after giving birth and the death of their pups”, entire section: Replace “We observed that pangolin offspring FG6 gave birth on 15 February 2018 and her pup died on 19 March. FG6 mated with males three times from 9-11 April and gave birth to the second litter of the year on 6 October; i.e. 53 days after giving birth and 22 days after the death of her pup, she successfully conceived. Another example was that the female captive born pangolin FG10 gave birth on July 26, 2018, and her pup died on the 29th. She was caged with male pangolin WM6 from 6-10 August. During the caging period, she had mated four times, and given birth on February 12, 2019 with a gestation period 186-189 days. Therefore, she mated and conceived just 11 days after calving and 8 days after the death of her pup. Another example was the female pangolin FG16 gave birth on August 26, 2019, and her pup died on 29th. She then mated with the male pangolin WM9 in cage four times from 3-5 September. She then gave birth on March 8, 2020 with a gestation period of 185-187 days. Therefore, FG16 had mated and conceived eight days after calving and five days after the death of her pup. The reason and mechanism for successful conception of young females in such short period of time after giving birth to and the death of pups is unknown. It may be a reproductive strategy to maintain and expand the population of this species.” with “Pangolin FG6 gave birth on 15 February 2018 and her pup died on 19 March 2018. She mated three times between 9–11 April 2018 and gave birth to a second pup on 6 October 2018. This indicates that she conceived again 53 days after giving birth and 22 days after the death of her pup. Female FG10 gave birth on July 26, 2018, and her pup died three days later. She was caged with male WM6 from 6–10 August, during which time she mated four times and gave birth on February 12, 2019 after a gestation period 186–189 days. Therefore, she mated and conceived just 11 days after pupping and eight days after the death of her pup. Another example is female FG16, which gave birth on August 26, 2019 and her pup died three days later. She was then placed with a male and mated four times between 3–5 September 2019. She gave birth on March 8, 2020 after a gestation period of 185–187 days. Therefore, FG16 had mated and conceived eight days after pupping and five days after the death of her pup. The reason and mechanism for successful conception of young female pangolins in such a short period of time after giving birth to and the death of pups is unknown. It may be a reproductive strategy to maintain and expand the population of this species.”

82. “The proportion of males with mating willingness is low”, Lines 2–3: Change “Of these pangolins, there was three of them (WM6, WM8 and WM9) showed strong willingness for mating” to “Of these pangolins, three (WM6, WM8 and WM9) showed a strong willingness to mate”.

83. “The proportion of males with mating willingness is low”, Line 4: Change “The male pangolin WM11” to “Male pangolin WM11”

84. “The proportion of males with mating willingness is low”, Lines 5–7: Change “Another male

pangolin WM12 mated with the female pangolin WF2 in April 2017, but did not show any mating after mating with the WF2 pangolin" to "Another male pangolin (WM12) mated with female WF2 in April 2017, but did not show any further mating activity"

85. "Captive pangolins reproduce more female than male offspring": Change this heading to "Captive pangolins produce more female than male offspring"

86. "Captive pangolins reproduce more female than male offspring", Line 1: Change "sexual ratio" to "sex ratio".

87. "Captive pangolins reproduce more female than male offspring", Line 2: Change "in our base" to "in our facility".

88. "Captive pangolins reproduce more female than male offspring", Lines 3-4: Change "conceived in wild" to "conceived in the wild".

89. "Captive pangolins reproduce more female than male offspring", Line 4: Change "with two offspring were twin" to "with two offspring being twins"

90. "Captive pangolins reproduce more female than male offspring", Lines 5-7: Change "Of these 44 offspring, the sex of 38 offspring were identified in time and the remaining failed to identify due to reasons such as offspring died in mothers' body without identifying their sexes" to "Of these 44 offspring, the sex of 38 were determined while the sex of the remaining individuals could not be determined as they died in vivo prior to being fully developed."

91. "Survival of wild pangolins and captive-born pangolin offspring", Line 2: Change "rescue base" to "rescue center".

92. "Survival of wild pangolins and captive-born pangolin offspring", Line 2: Delete "With our methods".

93. "Survival of wild pangolins and captive-born pangolin offspring", Line 7: Change "base" to "center".

94. "Survival of wild pangolins and captive-born pangolin offspring", Line 7: Change "survive" to "survived".

95. "Survival of wild pangolins and captive-born pangolin offspring", Lines 8-10: Remove these specific references to how long individuals survived – these data are available in your supplementary material. Rather include a summarized version of these data as you have done for captive-bred pangolins (i.e., X individuals survived >500 days, Y individuals survived >1000 days, etc.).

96. Figure 3, legend: Delete the sentence "High survival rates of wild pangolins and captive-born pangolins support the view that our methods are considerably good for the rescue and breeding of this mammalian species." This is an interpretation/opinion based on your graph, and therefore should not form a part of the figure legend.

97. "Survival of wild pangolins and captive-born pangolin offspring", Lines 22-24: Replace "Notably, we had first time bred 49 pangolins spanning three filial generations within the 5-year period. With our methods, twenty of them survive up to date (Supplementary Table 2)." with "Notably, during this 5-year period we had successfully bred 49 pangolins spanning three filial generations, 20 of which were still alive at the end of this study (Supplementary Table 2)."

98. "Survival of wild pangolins and captive-born pangolin offspring", Lines 27-29: Again, remove these specific references to the number of days that each individual survived – these data are available in your Supplementary Material Table 2. Rather just summarise it as "four individuals survived >900 days, while six individuals survived >1000 days".

99. "Survival of wild pangolins and captive-born pangolin offspring", Lines 30-31: You cannot make this statement, as you have not provided any information on how you "rescued" these individuals. Rescue goes far beyond maintenance, and includes all veterinary procedures and examinations, medication, etc. You can only say that you were able to maintain both wild-sourced and captive-bred individuals in captivity.

100. "Captive-born pangolins have better survival rates after weaning", Lines 5-6: "Notably, the 150 days of pre-weaning period was based on our latest research on captive-born pangolins manuscript in preparation)." You cannot say this, as this is an orphan sentence. Also, there is published literature that states that the weaning period lasts for 90-120 days (Chong et al. 2020. Sunda Pangolin species account, Chapter 6. In in Challender, D.W., Nash, H.C. and Waterman, C., 2019. Pangolins: Science, Society and Conservation. Academic Press). If you want to say this you first need to acknowledge the published literature, and then say that your own observations suggest that the weaning period in captivity is longer at 150 days.

101. Figure 4 caption: Change the caption to read: "Figure 4. Mortality of captive-born pangolins. (a) Mortality in captive-born offspring as a function of ages. (b) Proportion of surviving offspring

for two different groups (pre-weaning and post-weaning) across different years.

102. "Mortality and cause of death", Line 4. You cannot cite Clark (2008) here. You are reporting on your findings, and as Clark (2008) did not perform the autopsies and subsequently publish the results, you cannot cite her here. Instead, you should cite her in your Discussion when you mention that pyloric ulcers and bronchial complications are commonly seen in pangolin post-mortems.

103. "Mortality and cause of death", Lines 4-6. "Bacterial isolation of lung and liver tissues showed that most of the detected bacteria were known conditioned pathogens such as *Morgellons*, *Escherichia coli*, *Klebsiella pneumoniae*, and *Staphylococcus aureus*". This is the first time that you mentioned molecular characterization of bacteria isolated from lung tissue. Were these tissue samples collected during the post-mortems, and if so, how? Who did the molecular characterization and where? What primers did you use to amplify the pathogen DNA? What were the amplification conditions? How did you determine the identity of the isolated pathogens (e.g., via BLAST Search or phylogenetic inference)? All of this needs to be included in your Materials and Methods section.

104. "Mortality and cause of death", Line 1. How many are "some adult individuals"? You need to provide the specific number.

105. "Mortality and cause of death", Line 7. Change "were dissected" to "was dissected".

106. "Mortality and cause of death", Line 16. Change "the presumably" to "the presumed".

107. "Mortality and cause of death", Line 17. Change "suspected" to "suspect".

108. "Mortality and cause of death", Lines 18-19. Change "pangolins were infected" to "pangolins to be infected by".

109. "Mortality and cause of death", Line 19: Please provide the full reference for "Choo, Genome Research", or else clarify what you mean by this. Is this unpublished data?

110. "Mortality and cause of death", Line 20: Change "to make sure the food" to "to ensure that the food"

111. "Mortality and cause of death", Lines 22-23: Change "For instance, we increased the number of times to clean and wash cages (e.g. 1-2 times/month)" to "For instance, we increased the frequency with which cages were cleaned to 1-2 times per month"

112. "Mortality and cause of death", Line 23: Change "torches; moreover..." to "torches. Moreover..."

113. "Mortality and cause of death", Line 23: What do you mean by "moreover, we cleaned nests immediately when pangolins stop eating"? Does this mean that you cleaned the cages as soon as an individual stopped eating, or was its food removed when it stopped feeding?

114. "Duration of gestation in pangolins": change this heading to "Duration of gestation in Malayan pangolins"

115. "Duration of gestation in pangolins", Lines 2-4: Replace "When we put the female and male pangolins were placed in a cage for a period, they might mate several times and difficult to determine which mating was successful and lead to conception" with "When females and males were placed together in a cage, they might mate several times and it is therefore difficult to determine which mating was successful and led to conception"

116. "Duration of gestation in pangolins", Line 8: "with a gestation range of 154-203 days". This differs from the values that you present in Table 2.

117. "Duration of gestation in pangolins", Lines 8, 15 and 17: How could you determine these gestation periods if in most instances females mated multiple times, ostensibly also while they were pregnant (as reported in your Table 2)? For these values to be accurate, one needs to assume that the first mating always resulted in conception, but I have not seen any evidence to show that this is the case.

118. Figure 5 caption: Delete "Distribution of the gestation length."

119. Discussion, Line 2: Change "pangolins" to "Malayan pangolins"

120. Discussion, Line 2: Replace "Although previous study reported the breeding" with "Although previous studies have reported the breeding"

121. Discussion, Line 3: Replace "to first filial generation" with "to the first filial generation"

122. Discussion, Line 3: Delete "however"

123. Discussion, Line 4: "which we summarized in Supplementary Table (Zhang et al , 2015." Do you mean that you have summarized these differences in a Supplementary Table? If so, which Supplementary Table (provide the number/s). And if that is the case, why is the "Zhang et al. 2015" reference included here?

124. Discussion, Line 5: Delete the second "in".

125. Discussion, Lines 6 & 7: "since hundreds of years ago". This is incorrect, as this has only happened over the past ~150 years. Rather say "over the past two centuries".
126. Discussion, Lines 7 & 8: "very long period (e.g. up to 7 years)". 7 years is not a "very long period" for a species that probably lives for 20-30 years in the wild. If you must say this rather say that you have maintained them in captivity for "an extended period of time".
127. Discussion, Lines 10 & 11: "These advances have implications for conservation of endangered pangolins." If you are going to include this statement, you need to provide evidence on how your study has implications for the conservation of endangered pangolins. Breeding a species in captivity does not necessarily confer a conservation implication – it just means that we can breed them in captivity. For a captive breeding program to confer a conservation benefit it needs to consider such factors as genetic diversity, population genetics, diseases, geographical variation in morphology and genetics and adaptability of the captive-bred individuals to wild conditions, amongst various other factors. It is also pointless to be able to breed an endangered species in captivity if there are no safe areas into which these captive-bred individuals could realistically be released.
128. Discussion, Line 12: Add a comma after "diseases".
129. Discussion, Line 14: Replace "provide" with "provides"
130. Discussion, Line 32: Replace "observations was that the female pangolins could mate and conceived in very short" with "observations was that female pangolins could mate and conceive in a very short"
131. Discussion, Line 33: Replace "co-habiting" with "cohabiting"
132. Discussion, Lines 33-34: What is meant by "or the random mating in cages has a higher pregnancy rate"? A higher pregnancy rate than what?
133. Discussion, Line 34: "It is possible that pangolins are solitary animals". This is not a possibility, it is a fact and is mentioned in nearly every article dealing with pangolins.
134. Discussion, Lines 36-37: Replace "In order to breed species, the placental mammal might evolve to have efficient breeding strategies." with "In order to breed, placental mammals have evolved various breeding strategies."
135. Discussion, Lines 43-45: "In other rescue projects of Malayan pangolins in China,
136. it has been understood that the proportion of wild male Malayan pangolin willing to
137. mate is also relatively low." You need to provide a reference for this statement. Which other rescue centers have reported a low willingness to mate among male pangolins?
138. Discussion, Line 51: The correct scientific name for a Jaguarundi is *Herpailurus yagouaroundi*.
139. Discussion, Line 52: The correct scientific name for a lion is *Panthera leo*.
140. Discussion, Line 52: The correct scientific name for a blackbuck is *Antelope cervicapra*.
141. Discussion, Line 53: The correct scientific name for an amur leopard is *Panthera pardus orientalis*.
142. Discussion, Line 53: The correct common name for a pudu is a southern pudu.
143. Discussion, Line 54: "R.Glatston 1997". Why include the author's initial?
144. Discussion, Lines 55-57: Change "Faust and Thompson analyzed the sex ratio of 66 captive mammalian species, they only found two species showing female-biased sex ratios (Faust and Thompson 2000)." to "Faust and Thompson (2000) analyzed the sex ratios of 66 captive mammalian species and found only two species that showed female-biased sex ratios."
145. Discussion, Line 58: The correct scientific name for the pygmy hippo is *Choeropsis liberiensis*.
146. Discussion, Line 73: Change "are still rooms for improvement" to "is still room for improvement".
147. Discussion, Lines 73-75: Replace "First, it would be interested in future to evaluate the genetic structure of these captive pangolins and compared with wild pangolins" with "Firstly, it would be interesting to evaluate the genetic structure of these captive pangolins and compare this with wild pangolins"
148. Discussion, Lines 75-76: "although it may be difficult to get the endangered wild pangolins for analysis". There is a wealth of genetic data for wild pangolins already, so this statement is incorrect and should be removed.
149. Discussion, Line 77: Replace "planning to look for more male pangolins into our base" with "planning to source additional male pangolins for our center".
150. Discussion, Line 78: Replace "preweaning" with "pre-weaning".
151. Discussion, Lines 79-80: "Fourth, we will train and gradually release captive-bred pangolins back to the wild.". This is not a good idea unless it follows the IUCN guidelines for reintroductions.

As mentioned above, any reintroduction program is pointless unless there are areas that are secure enough for pangolins to be released into, where there isn't an existing pangolin population. Furthermore, any reintroduction program needs to consider genetic diversity, population genetics, diseases, geographical variation in morphology and genetics and adaptability of the captive-bred individuals to wild conditions, amongst various other factors.

152. References: Reference no. 3 (Challender et al., 2019). This is not the complete reference and several author's names have been omitted from this reference. Please correct this.

153. References: "R.Glatston". Why does one of this author's initials precede his family name?

154. The references require specific attention. In some instances, the authors initials and surname are provided, while in other instances the authors' entire names and surnames are provided. In at least two instances (Challender et al. 2019 and Heinrich et al. 2016) the authors are incorrectly indicated.

Reviewer #3 (Remarks to the Author):

I enjoyed reading this article, and while I think it will make a contribution to the literature, there are a number of areas of the manuscript that require major work if it is to be published in a good quality international journal. I expand on these points below.

*I appreciate that English may not be the first language of the authors. However, I would encourage them to check the use of English, or ask an English-speaking colleague to do so, so that they hard work that has gone into this research is clearly conveyed to the reader.

*What do the authors mean by systematic in terms of their research? How was the breeding systematic?

*The paper is largely descriptive and would benefit further linking of the results presented to existing knowledge of pangolin husbandry and care, and life history. E.g., why might there be a low willingness to mate among males? What were the causes of death of animals that did not survive in this study? The authors say 'keeper management' but this is not very specific -what is actually meant by such terms? Many of the points made are superficial and not explored in very much depth at all which is disappointing. More detailed thinking here is needed and these such points should be elaborated on.

*Can the authors clarify what is happening with weaning? Fig 4b suggests that once an animal is weaned it survives. But there has been a decline in weaning success between 2016 and 2020. Why is this?

Minor points:

*Title - check the syntax

*L19/50 - change to "threatened"; same for introduction

*L22 mating and reproduction?

*L28. How are you defining conception rate? Please make this clear for the reader.

*L36 - how do this captive rate relate to wild populations?

*L40/80 - can 5 years (less than the lifespan of *M. javanica*) be considered long term? Suggest revising.

*L46 - the authors should suggest additional key words.

*53-55 - Check house rules - should you include scientific names?

*58 - Suggest you refer to the most up to date version of the IUCN Red List, which is 2021.

*60 - there are better references than NGO report- see here:

<https://www.sciencedirect.com/science/article/pii/B9780128155073000034>

*74 - Chinese pangolins are now living into their mid-20s in Taipei Zoo - see here:

<https://www.sciencedirect.com/science/article/pii/B9780128155073000368>

*78 - this is a more up to date reference -

<https://www.sciencedirect.com/science/article/pii/B9780128155073000289>.

*L80 - what is 'very successful'?

*L110 - So there was case of twins being born? Please clarify in the text.

*L23-124 - was the floor wire mesh?

*L136-138 - in what quantities were each of these items provided?

*Figure 2 is a nice figure.

*The first cage mating has a high conception rate: please explain how first and second caging

worked – how many days were there between the first and second caging?

*Please elaborate on why you think the conception rate is high based on knowledge of the species' ecology and biology.

*Why do you think *M. javanica* copulates while pregnant?

*Any evidence of females breeding while they are rearing young? This has been reported in wild Chinese pangolins.

*How did you determine willingness to mate among males? Please provide further details.

*Please elaborate on why you think there is a low/no willingness to mate among some males? Are there theoretical and/or practical reasons why this might be the case?

*Fig. 3. What were the strict measures to improve survival rates? Please elaborate on any changes to care given.

*Of the animals that did not survive beyond e.g., 150 days – what was the cause of their death?

*Can the authors elaborate on what are the major problems with weaning? This still seems to be causing problems with breeding and rearing *M. javanica*.

*On deaths – are the respiratory diseases related to not having enough room in their cages (which seem small) and which could therefore affect lung function?

*What are gas torches? Please clarify for the reader.

*Change daughters and sons to males and females.

Response to Reviewers

Reviewers' Comments to the Authors:

Reviewer #1:

In this manuscript, Dingyu et al. describe the establishment of a self-sustaining captive pangolin population. This is exciting because pangolins have been generally difficult to maintain in captivity, despite the increasing need to support wild populations. As this manuscript is descriptive in nature, I have no issues with the design of the project. However, I do think a few things require a bit more explanation.

From the many descriptions in the text, these pangolins will readily mate and produce offspring in the captive environment. What was done differently here that made this captive population work compared to all the previously failed attempts? I understand that the strict cleaning and sterilization measures implemented in 2019 increased survival after that time, but what led to the increased survival relative to other programs before 2019? More explanation of the differences between other and this project are needed to really understand what caused this so the techniques could be applied to other populations and species.

Author response: Thank you for your suggestions! Based on our experience, we believe that it could be related to our practice or experimental design that provide a hygienic environment to pangolins in captivity. For instance, we used artificial food instead of providing ants/termites that could be sources of pathogens. Moreover, the pangolins were kept in cages that were equipped with wire mesh that help to clean their bodies. We also clean and sterilise the cages, especially when they were sick. In 2018, we found that many pangolins were still infected because our measures were not strict enough. For example, our artificial food did not keep it in a proper place and caused the growth of pathogens in the food. Therefore, we implemented more strict measures by placing the food in a dry environment to minimise the growth of pathogens in the food. Besides that, we increase immediately sterilise the cages when pangolins had no appetite to eat. These strict measures have significantly increased the survival of pangolins. We have mentioned the strict measures in the revised manuscript (Lines 619–627).

It is interesting that even in only 2 generations, survival of captive individuals is less than survival of wild individuals in captivity. Typically, I think of survival in captivity as being equal among captive and wild born individuals or perhaps favoring captive born individuals via adaptation to the captive environment over time. Do you have some explanation (or hypothesis) that can explain why captive born individuals did not survive as well as wild born individuals in captivity?

Author response: Of the seven pangolins born in 2016, five were born after their mothers conceived in the wild and admitted to our centre. Four of them survived to adulthood, and one survived for 154 days. The other two females mated at the research centre and survived to adulthood. Therefore, the pups conceived in captivity have similar or even higher survival rate as those conceived in the wild.

However, the survival rate of our captive-born pangolins at preweaning was considerably low, which also observed in other captive animals. Since most of the wild pangolins that we brought into our centre were adults, we are unsure whether the survival rate of the wild pangolins at preweaning is low or high as the data is unavailable.

Finally, I understand that the mating pairs were random because you lacked any existing data on the relatedness between most wild-born individuals. However, it isn't entirely clear how the available information was used. For example, lines 110-111 say that captive born males and females were used for mating; were these full or half siblings? In addition, you can now make some calculations about founder genome equivalents that are present in your current captive population and use these to lay out your plans for managing this population moving forward and provide some hints as to the relationship of the captive born individuals. For example, how many individuals are full or half siblings? Will you initiate a minimising mean kinship approach for selecting mates in future generations, now that you know parentage? How many males will you need to add to your population to reach your desired population size while maintaining genetic diversity? Adding these details will help show that you have indeed produced a self-sustaining pangolin captive population.

Author response: Sorry for the confusion. The “random” here does not mean we randomly selected pangolins for mating. It actually means we selected pangolins for mating with random dates, or the date of cohabitation is irregular because the oestrus of female pangolins is unknown. We have clarified this point in the revised manuscript (Lines 213-214).

The selection of pangolins for mating was not random, as we avoided inbreeding in our experimental design. All pangolins were assigned with unique IDs, and their relationships (e.g., siblings) were recorded. When selecting pangolins for breeding, we would avoid inbreeding based on this recorded information (only two out of 48 pregnancies were inbreeding because we would like to explore the possibility and behaviour of inbreeding in pangolins). We have clarified this point in the revised manuscript. (Lines 213-216).

For the 33 wild pangolins that we initially brought into our centre, we believe they are unlikely to be close relatives because they were received into the centre in 11 independent batches spanning four years. There is little possibility that they are close relatives. We will, of course, conduct genetic analysis of them in the future to make more concrete conclusions. We have used more appropriate words and added clearer explanations in our revised manuscript.

Minor comments:

Lines 19-20: Something seems off grammatically with this sentence.

Author response: Addressed.

Line 75: Should “to farming” be removed? If not, this needs more explanation.

Author response: Yes, we agree. We removed “to farming”. (Line 103)

Line 84: I think this should be, “Here, for the first time, we report a large-scale...”

Author response: We removed ‘large-scale’ and ‘long-term’ and rephrase the sentence. (Line 110).

Line 84: I do not think that 5 years/3 generations is “long-term” please consider rephrasing this.

Author response: We have removed it in the revised manuscript. (Line 110)

Line 105: How were confiscated pangolins distributed across China? Were they all from the same locality?

Author response: For your information, the confiscated Malayan pangolins were originated from Southeast Asia (Lines 90–91). They were from several independent confiscations by authority at different dates and places in China. These pangolins were confiscated from border across China and Vietnam. Therefore, they all are unlikely from the same locality.

Line 108: Were the 5 litters born in the field considered wild born and were they ever mated together? I know the mating was random but it seems that sibling pairs should be avoided.

Author response: Sorry for the confusion. The five pups were born between 79–136 days after their mother’s pangolins arrived in our centre. They were not used to mate with any other male pangolins at our centre before giving birth to pups. Yes, we did not select sibling pairs for mating (except two out of 48 pregnancies were intentionally selected for the study of inbreeding in pangolins). Again, by “random”, we mean the date of cohabitation that we choose is irregular and not randomly selected pangolins for cohabitation. We have corrected this in our latest manuscript. (Line 214).

Line 145: Later on you explain what this training is but it is confusing here. I suggest you add some language that you will expose pangolins to more natural condition prior to releasing them back into the wild.

Author response: We have revised it as suggested. (Lines 735–747)

“we bred the offspring and calculated its successful rate” I am not completely sure what

you mean here, maybe 'success rate' instead of 'its successful rate'?

Author response: This part has been deleted in the revised manuscript. (Line 296)

"August 2017 to November2020" missing a space between November and 2020.

Author response: Addressed. (Line 300)

"For instance, FG3 and FG10 bred three offspring, FG6 and FG16 bred two offspring, and the other five captive-born pangolins bred an offspring." I am not familiar with this kind of use of the word 'bred'. In this sentence I would use 'produced' instead.

Author response: Thank you and we appreciate your suggestions. This part has been deleted in the revised manuscript. (Lines 301–303)

Reviewer #2:

This manuscript reports on the breeding success of 11 wild-caught Sunda pangolins (*Manis javanica*) in captivity, together with the breeding success of their captive-born offspring. Although previous studies have investigated captive breeding success, this is the first study that I know of that has done so in a systematic way and has reported successful breeding of the F2 generation.

However I found this manuscript overall to be fairly poorly written, and often difficult to follow. In your Introduction, you need to introduce your study species in more detail, specifically including details such as published average and maximum masses and lengths, and published gestation periods. This will allow readers to better interpret your results.

Author response: Sorry for the overall English language quality. We have polished the language in the revised manuscript. As suggested by the reviewer, we have added this useful information in the introduction. (Lines 112–116)

I found the Methodology to be confusing. For example, I am unsure whether male and female pangolins were housed separately on two ends of a cage with an intervening 'common area', or whether they were placed together in a single cage during copulation. In your results section you also mention that bacterial diversity in the lung tissue of

pangolins that had died was investigated using molecular techniques, but this is not mentioned at all in your Methodology.

Author response: Sorry for the confusion. For the cage structure, each pangolin has a cage that is separated into three parts with two sections on one side for sleeping in winter and summer, and the rest part on the other side is the activity area. Two cages have been placed in a line, and the two activity areas were adjacent to each other with a door in between. We open the door whenever we choose the date for them to cohabit, which you can see in the mating video. We have made it clearer in the revised manuscript (Line 216–220). Besides that, we have added the method description of molecular techniques in the revised manuscript (Lines 252–256) and Supplementary Information file.

I also found the Results section difficult to follow, mainly because the results are often analysed and discussed directly after being mentioned.

I think that the quality of this manuscript would be greatly enhanced if the results are separated and treated discretely in the Results and Discussion sections, i.e., use the Results section just to report your Results, and use the Discussion to analyse and compare these results to previous studies.

Author response: Thank you for pointing this out. We have moved the discussions in the Results section to the Discussion section.

Please find below a list of specific queries, concerns and comments that I have:

1. Throughout the manuscript, use an en-dash (not a dash) to indicate a range of values.

Author response: Thank you for pointing this out. We have made this change in our revised manuscript.

2. Line 1: Change the title to “Systematic captive breeding of Malayan pangolins: implications for conservation”.

Author response: We have made this change in our revised manuscript. (Lines 1–3)

3. Line 19: Replace “endangered” with “threatened”. “Endangered has a very specific meaning in conservation, as it refers to a category on the IUCN Red List. As the eight pangolin species are variously classified as Vulnerable, Endangered and Critically Endangered on the IUCN Red List, using the word “endangered” in this context infers a false meaning.

Author response: Thank you for pointing this out. We have replaced it in the revised manuscript. (Line 31)

4. Lines 19-20: Replace “carried out to 20 captive pangolins since hundred years ago” with “undertaken in the last century to maintain pangolins in captivity”.

Author response: We have made this change in the revised manuscript. (Lines 32–33)

5. Line 22: Replace “mating reproduction” with “breeding”.

Author response: We have made this change in the revised manuscript. (Line 36)

6. Lines 25-26: Replace “10 of them successfully produced” with “10 of successfully produced”.

Author response: We have deleted this part in the revised manuscript. (Line 41)

7. Line 34: Replace “9-month” with “9-months”.

Author response: For consistency, we have changed it to “nine months” in the revised manuscript. (Line 51)

8. Line 35: Replace “captive pangolin” with “captive pangolins”.

Author response: We have deleted this part in the revised manuscript. (Lines 52–53)

9. Lines 38-39: Replace “have similar gestation length” with “have a similar gestation length”.

Author response: We have made this change in the revised manuscript. (Lines 57–58)

10. Line 39: Replace “with the wild pangolins.” with “to the wild-caught pangolins.”.

Author response: We have made this change in the revised manuscript. (Line 58)

11. Lines 50-51: Replace “Pangolins are the most endangered wildlife species and recognised as the most 51 ravaged mammal by illegal trade” with “Pangolins are the most trafficked group of wild mammals”.

Author response: We have made this change in the revised manuscript. (Line 85)

12. Lines 52-53: Replace “There are eight modern pangolin species” with “There are eight extant pangolin species”.

Author response: We have made this change in the revised manuscript. (Line 82)

13. Lines 53-55. These common names are very outdated. Use the most recent common names as accepted by the IUCN and published on the IUCN Red List, and also include the most recent scientific name for each species after the common name.

Author response: We have revised it as suggested. (Lines 83–85)

14. The in-line referencing needs to be corrected. For example, in some instances the author’s surname and year are provided, but in other instances (for example line 56) the authors’ full name, abbreviated second name and surname are provided, in addition to the year (“Timothy J. Gaudin 2009”).

Author response: We have revised it as suggested.

15. Line 57: Why mention where Chinese pangolins occur? You did not use them in your experiment, nor did you mention where the remaining six species occur. Remove this section.

Author response: We have removed it in the revised manuscript. (Lines 78–79)

16. Lines 56-57. Include additional information on the Malayan pangolin, specifically what is the average and maximum mass, and the average and maximum body length? This will

help readers to determine whether the individuals that you used in your experiments were juveniles, young adults or mature adults. Also include what is currently known about their gestation and weaning periods.

Author response: Agreed. We have included this information in our revised manuscript. (Lines 112–116)

17. Line 58: This reference is very outdated. The most recent IUCN conservation assessment is from 2019 – use that.

Author response: We have revised it as suggested. (Lines 781–785)

18. Line 63: Merge these references to read: “Choo et al. 2016, 2020; Challender et al. 2019”.

Author response: We have revised it as suggested. (Line 86)

19. Line 64: Change “wildlife” to “wildlife species”.

Author response: We have deleted this part in the revised manuscript. (Line 73)

20. Line 65: Change “extinction” to “continued existence”.

Author response: We have deleted this part in the revised manuscript. (Line 96)

21. Line 74: Additional references that you should include are Hua et al. 2015, and Hoyt, R. 1987. Pangolins: Past, present and future. Proceedings of the AAZPA National Conference: 107-134.

Author response: We have revised it as suggested. (Line 124)

22. Line 75; Replace “farming” with “captive maintenance”. None of the references that you cite here specifically looked at the challenges to farming pangolins. Rather, they all considered the challenges to maintaining pangolins in a captive environment.

Author response: We have deleted this part in the revised manuscript. (Line 103)

23. Lines 77-78. Again, in-line references need to be standardised. Here the author's initial is sometimes given, while at other times it is not.

Author response: We have checked these issues in the revised manuscript.

24. Lines 78-81. This sentence needs to be reworded, as in its current state it is very difficult to understand. Also, in line 81, combine these references. I suggest changing it to: "Although there have been isolated reports of pangolins breeding in captivity, these offspring were born from a small number of wild-caught adults and these offspring failed to sire F2 generation offspring after having been reared in captivity (Zhang et al. 2015, 2017)."

Author response: We have deleted this part in the revised manuscript. (Lines 106–109)

Lines 82-83: Again, this sentence is unclear and needs to be reworded. I suggest changing it to: "Therefore, breeding pangolins in captivity remains challenging and as a result they are one of the most difficult mammals to breed in captivity globally".

Author response: As suggested, we have revised it in the revised manuscript. (Lines 131–133)

25. Lines 84-85 need to be reworded. I suggest "We report on the first large-scale and long-term captive breeding program for critically endangered Malayan pangolins spanning a 5-year period from 2016 to 2020."

Author response: To make it brief, we have revised it to "The current study reports the first successful captive breeding programme for the critically endangered Malayan pangolins" in the revised manuscript. (Lines 134–135)

26. Lines 85-93 (starting at "Our data..."). These are Results and Discussions and should not be included in your Introduction. Remove this section.

Author response: We have deleted this part in the revised manuscript. (Lines 135–141)

27. Lines 104-106. Reword to: All 33 Malayan pangolins used in this study were of wild

origin and were confiscated by law enforcement officers from smugglers in China (Supplementary Table 1).”

Author response: We have revised this sentence. (Lines 155–157)

28. Lines 106-111. Replace “litter” with “offspring”.

Author response: We have deleted this part in the revised manuscript. (Lines 158)

29. Lines 106-111 needs to be reworded. I suggest: “By November 30, 2020, we had 48 offspring born at our rescue centre at Guangxi Academy of Forestry Science, of which 46 were normal births (Supplementary Table 2). Among of these, five offspring were conceived in the wild but were born at our centre. The remaining 43 offspring were conceived at our centre, and of these 41 were normal births, with a single instance of twins being born. Eleven wild females, three wild males, twelve captive-born females and a captive-born second-generation male were used for mating (Supplementary Table 3).”

Author response: We have deleted this part in the revised manuscript. (Lines 1587–161)

30. Line 112: Change “large scale and long-term experiments” to “large-scale and long-term experiment”.

Author response: We have deleted this part in the revised manuscript. (Line 110)

31. Line 113: Change “base” to “centre”.

Author response: We have made this change in the revised manuscript. (Line 167)

32. Line 116: Change “are originated” to “originate”.

Author response: We have made this change in the revised manuscript. (Line 180)

33. Line 117. Capitalise “Indonesia”.

Author response: We have made this change in the revised manuscript. (Line 180)

34. Line 117: Change “climate” to “climates”.

Author response: We have made this change in the revised manuscript. (Line 181)

35. Line 118: Add a comma after “zone”.

Author response: This part has been deleted in the revised manuscript. (Line 172)

36. Line 119: Change “the indoor cage” to “indoor cages”.

Author response: We have revised it as suggested. (Line 173)

37. Line 128: Delete “of pangolins”.

Author response: Deleted.

38. Line 129: Change “setting” to “settings”.

Author response: We have made this change. (Line 188)

39. Line 138: Change “fodder” to “food”.

Author response: We have changed “fodder” to “formula”. (Line 200)

40. Line 139: Change “fluid” to “liquid”.

Author response: We used “semifluid”, which may be more appropriate. (Line 201)

41. Lines 140-141: Change “Although the food had liquid, clean water was still provided in a separate bowl because some pangolins might need to drink it.” to “Although the food was liquid, clean water was still provided ad lib in a separate bowl.”

Author response: We have revised this sentence in the revised manuscript. (Line 203)

42. Line 142: Change “all food were stored at” to “all food was stored in at”.

Author response: We have made this change in the revised manuscript. (Lines 203–204)

43. Lines 144-145: Change “At the end of this project, all living pangolins are maintained

in our base for a long term observations and study.” to “At the end of this project, all living pangolins continued to be maintained in our centre for a long-term observational study.”

Author response: We have made this change in the revised manuscript. (Lines 206–207)

44. Lines 145-146. What do you mean by “Some of them will be subjected for training before we release them into the wild.”? Do you mean that you will be training some of these individuals, or will they be used to teach/train people? If the latter, what type of teaching will they be used for?

Author response: We have revised this section in the latest manuscript in order to make it clearer. (Lines 207–209)

45. Lines 151-153: Change “The doors of cages kept open and monitor was placed towards to the common room that they both stayed in order for continuously recording for 24 hours.” to “The cage doors were kept open and a monitor was placed in the common room that they both shared in order to continuously record their activity for 24 hours.”

Author response: We have revised the sentence. (Lines 216–220)

46. Lines 151-153. This sentence needs to be clarified. The way it is described is sounds like there are three cages: one cage on either side housing the two individuals separately, and a central “common room” that both individuals have access to once their respective cage doors are opened. Is this the case? Or were both individuals placed in a single cage?

Author response: Sorry for the confusion. Each pangolin has a cage separated into three parts with two sections on one side for sleeping in winter and summer, and the rest part on the other side is an activity area. Two cages have been placed in a line, and the two activity areas were adjacent to each other with a door in between. We will open the door whenever we choose the date for them to cohabit. (Lines 216–218)

47. Lines 153-154: Change “Whether the mating was successfully done or not were

determined according to their mating actions.” to “Whether mating successfully occurred was determined by their mating behaviour.”

Author response: We have revised it as suggested. (Lines 220–222)

48. Line 155: Change “the number of mating” to “the number of mating events”.

Author response: We have made this change in the revised manuscript. (Line 223)

49. Line 156: Change “date of calving and the pregnancy period” to “date of pupping and the gestation period”

Author response: We have replaced it with “date of delivery, and the gestation period”. (Lines 223–224)

50. Line 158: What do you mean by “early-born offspring”?

Author response: Sorry for the confusion. “Early-born offspring” means the offspring born in the early period (i.e., 2016–2017) of the start of this breeding programme. We have revised it to make it clearer. (Line 226)

51. Line 159: Change “measured for their head and tail lengths” to “their total length measured”.

Author response: We have made this change. (Lines 227–228)

52. Line 159: How was their total length measured? Was it measured while they were curled up, or while they were stretched out straight? These two methods result in different measurements, so the method used needs to be stated.

Author response: Thank you for pointing this out. As shown in Figure 2, we measured the total length while they were stretched out.

53. Line 173: Change “environment” to “environments”.

Author response: We have made this change. (Line 262)

54. Line 177: Change “the base” to “our centre”.

Author response: We have made this change. (Line 265)

55. Lines 179-180: “our data showed that the rate of mating and conception was generally high”. Where are these data? If you have presented them, then you need to include a reference to the table or figure. What do you mean by high? You need to include values here like how many individuals mated in how many experiments, and of those mated how many led to pregnancy? I assume that these data are all captured in Table 1, but you need to summarise them here as well so that the reader does not have to try and infer this information from your Table 1.

Author response: Thank you for pointing this out! We have included the required information in the revised manuscript. (Lines 269-277 and Lines 331-337)

56. Lines 182-183: As mentioned above, you cannot make this statement unless you include data on the actual numbers of attempted matings, successful matings and the proportion of individuals successfully conceiving and giving birth.

Author response: Please refer to the response above.

57. Figure 1a. I do not understand why this figure or the associated video have been included. It is a well-known fact that pangolins carry their offspring on their tails (see for example introductory chapters and species accounts in Challender, D.W., Nash, H.C. and Waterman, C., 2019. Pangolins: Science, Society and Conservation. Academic Press, and references therein).

Author response: Sorry for the confusion. Figure 1 aims to demonstrate the situation of the pangolins at our centre. We showed the mother pangolin carry her offspring because many readers (particularly those are not pangolin experts) may not be aware of this fact.

58. “Captive-born female pangolins have good reproductive success”, Line 2: Change “in artificial environment” to “in an artificial environment”.

Author response: We have changed it to “under the controlled environment”. (Lines 295–296)

59. “Captive-born female pangolins have good reproductive success”, Lines 2-3: Change “calculated its successful rate” to “calculated the success rate”.

Author response: This part has been deleted in the revised manuscript. (Line 296)

60. “Captive-born female pangolins have good reproductive success”, Lines 3-6: This section is confusing. In lines 3 & 4 you say that there were 12 F1 offspring, and that these produced 18 viable F2 offspring. But in lines 5 & 6 you say that there were 10 F1 offspring, and that they produced 15 F2 offspring. Are lines 5 & 6 supposed to refer to F2 (i.e., second generation) offspring?

Author response: We have revised it in the latest manuscript. (Lines 297–300)

61. “Captive-born female pangolins have good reproductive success”, Line 5: Change “November2020” to “November 2020”.

Author response: We have revised it in the revised manuscript. (Line 300)

62. “Captive-born female pangolins have good reproductive success”, Lines 6-8: Remove the sentence starting “For instance,…” and ending “bred an offspring”. These details are captured in Table 1, and do not provide any additional information that needs to be repeated in the text.

Author response: We have deleted this part in the revised manuscript. (Line 301)

63. “Captive-born female pangolins have good reproductive success”, Line 12: What do you mean by “artificial breeding”? Artificial breeding means the artificial insemination of a female using sperm that was harvested from a male. This has not been mentioned at all in this manuscript, and I therefore assume that “artificial breeding” is used in error here.

Author response: Sorry for the confusion and thank you for pointing this out. We have deleted this part in the revised manuscript. (Line 307)

64. “Captive-born female pangolins have good reproductive success”, Lines 13 & 14: You cannot say that your diet and conditions are “suitable for the normal growth of these captive pangolins” as nowhere have you compared the growth rates of these captive-born pangolins to the growth rates of wild-born pangolins. At best you can say that your conditions are suitable for rearing pangolins.

Author response: This part has been deleted in the revised manuscript (Lines 308–309)

65. Figure 2: Remove “Photograph” from all your captions. i.e., Caption 1 should just read “On October 28 2016,....”, Caption 2 should read “On February 7 2017, ...”, etc.

Author response: Agreed. We have revised the Figure 2. (Line 339)

66. Figure 2: Your caption “A characteristic white tail, a few dark scales among white scales” does not add any value as it in no way addresses the topic of your study, i.e. breeding pangolins in captivity. Remove this caption.

Author response: As suggested, we have deleted it. (Line 339)

67. Figure 2 legend. Change your current legend to “Figure 2. The growth process of the first-generation offspring FG6. Each captive-born pangolin was regularly weighed and measured (Photos by Dingyu Yan).” As mentioned above, your statement “Some unique features of pangolins were also highlighted.” adds no value as it has nothing to do with breeding pangolins in captivity. Also as mentioned above, you cannot say that “FG6 grew normally” as you did not compare her growth rates to those of wild pangolins.

Author response: We have modified this part in the revised manuscript. (Lines 339–341)

68. “The first cage mating has a high conception rate”, line 3: Change “produced” to “conceived”.

Author response: This part has been deleted in the revised manuscript. (Line 322)

69. “The first cage mating has a high conception rate”, line 9: Change “need” to “needs”.

Author response: This part has been deleted in the revised manuscript.. (Line 328)

70. Table 2: Change “Times of Mating” to “Number of mating instances”.

Author response: As suggested, we have changed it. (Line 345)

71. “Captive-born female pangolins reach sexual maturity in 7-8 months”, Lines 3-5: Replace “For instance, the second-generation SG4 had mated and conceived at 7th month after her birth, which is the earliest sexual maturity that we observed for second-generation pangolins.” with “The second-generation SG4 mated and conceived 7 months after her birth, which is the earliest sexual maturity that we observed for second-generation pangolins.”

Author response: We have revised this in the revised manuscript. (Lines 350–353)

72. “Captive-born female pangolins reach sexual maturity in 7-8 months”, Lines 5-7: Replace “For female pangolins FG10, FG15, SG4, and FG16, they were first conceived at 7-9 month old, even before the separation from their mothers.” With “Female pangolins FG10, FG15, SG4, and FG16 first conceived when they were 7–9 month old, even before being weaned from their mothers.”

Author response: We have modified it in the revised manuscript. (Lines 353–355)

73. “Captive-born female pangolins reach sexual maturity in 7-8 months”, Lines 6-7: Why were these young pangolins (7-9 months old) still with their mothers? I assume that this is because they were not able to disperse naturally because of their confined conditions? Various authors (e.g., Lim & Ng, 2008; Nguyen et al. 2014) report that weaning occurs at 3-4 months, suggesting that these offspring should have already been independent of their mothers. [Lim, N.T. and Ng, P.K., 2008. Home range, activity cycle and natal den usage of a female Sunda pangolin *Manis javanica* (Mammalia: Pholidota) in Singapore. *Endangered Species Research*, 4(1-2), pp.233-240. and Nguyen, V.T., Clark, V.L., Tran, Q.P., 2014. Sunda Pangolin (*Manis javanica*) Husbandry Guidelines. *Carnivore and Pangolin Conservation Program, Save Vietnam’s Wildlife, Vietnam.*

Author response: We did not separate them because we would like to further investigate on the behaviours between offspring and mothers.

74. “Captive-born female pangolins reach sexual maturity in 7-8 months”, Line 9: Change “the age of first conception was occurred slightly late between 11 to 18 months” to “the age of first conception occurred slightly later between 11–18 months”.

Author response: We have revised this in the revised manuscript. (Lines 356–357)

75. “Captive-born female pangolins reach sexual maturity in 7-8 months”, Lines 11-13: Change “as earliest as approximately 7-8 months after their births. Zhang, et al. (2015) inferred that sexual maturity of the Malayan pangolin occurred at approximately 1 year, even it could occur as early as 6–7 months” to “as early as 7 months of age. Zhang et al. (2015) inferred that sexual maturity of the Malayan pangolin occurred at approximately 1 year, although also reporting an instance of a captive female being pregnant at 6–7 months of age”

Author response: This part has been deleted in the revised manuscript. . (Lines 361–362)

76. “Female pangolins accept mating with male pangolins during pregnancy”, Lines 3-5: These data do not match what is in Table 1. For instance, Table 1 says that she was born on 22 August 2016 (not 6 October 2018).

Author response: It means FG6 itself was born on 22 August 2016, and FG6 gave birth to its pup on 6 October 2018. We have revised it in the latest manuscript. (Lines 367–368)

77. “Female pangolins accept mating with male pangolins during pregnancy”, Line 3: Change “For instance, the pangolin FG6 produced on 6 October, 2018 and mated with male four times (in five times of caging) before calving. Among of these four mating, one of them (9-11 April 2018) led to conception, whereas the reaming three mating occurred during pregnancy. The last mating occurred 33 days before calving with the male pangolin WM9. Another example was FG4 mated with WM8 seven times in 26-29 days before calving (from 6 to 9 March 2018) (Table 2). Therefore, we estimated that the two females

accepted to mate 53–32 days before calving. This phenomenon of mating during pregnancy also exists in wild female pangolins (Table 1).” to “For instance, pangolin FG6 pupped on 6 October 2018 after having mated four times (during five caging events). Among these four matings, one of them (9–11 April 2018) led to conception, whereas the remaining three matings occurred during pregnancy. The last mating occurred 33 days before pupping. Another example was FG4, which mated with WM8 seven times in 26–29 days before pupping (between 6 and 9 March 2018; Table 2). Therefore, we estimate that the two females accepted matings 53–32 days before pupping. This phenomenon of mating during pregnancy also exists in wild female pangolins (Table 1).”

Author response: We have revised it in the revised manuscript. (Lines 367–376)

78. “Female pangolins accept mating with male pangolins during pregnancy”, Line 6: Change “reaming” to “remaining”.

Author response: We have modified accordingly in the revised manuscript. (Line 370)

79. “Female pangolins accept mating with male pangolins during pregnancy”, Lines 10-11: “This phenomenon of mating during pregnancy also exists in wild female pangolins”. What is the reference for this statement? If you say that females in the wild also mate while they are pregnant, you need to provide evidence or a reference to support this.

Author response: Sorry for the confusion. Perhaps, the phenomenon exists in our confiscated wild female pangolins. We have revised the sentence to make it clearer in the revised manuscript. (Lines 375–376)

80. Table 1, 2 and throughout the manuscript: If females were placed with males multiple times before parturition, how do you know that they always conceived during the first pairing? Why could it not have been during the second or subsequent pairings? If you were using gestation period to determine likely date of conception, then you cannot report on gestation period here as that then becomes a circular argument. How do you know that gestation period isn’t actually a lot shorter than what you record, suggesting that females only became impregnated during a subsequent mating event?

Author response: Thank you very much for the good question. We observed that ten pregnancies of females were born when the females had only one cohabitation, and the gestation period was in the range of 178–193 days. We then define this range as the basic gestation period of Malayan pangolin. For example, a female cohabited twice (from 31 December 2019 and 3 March 2020) before parturition and gave birth to a pup on 1 July 2020. If the first cohabitation resulted in conception, the gestation period was 183 days, whereas it was 120 days if the second cohabitation resulted in conception. Since the 183 days is within the observed basic gestation period, we would consider the first cohabitation resulted in conception with a gestation period of 183 days. To make it clearer, we have explained this in the revised manuscript (Lines 238–250)

81. “Female pangolins conceive within a short period after giving birth and the death of their pups”, entire section: Replace “We observed that pangolin offspring FG6 gave birth on 15 February 2018 and her pup died on 19 March. FG6 mated with males three times from 9-11 April and gave birth to the second litter of the year on 6 October; i.e. 53 days after giving birth and 22 days after the death of her pup, she successfully conceived. Another example was that the female captive born pangolin FG10 gave birth on July 26, 2018, and her pup died on the 29th. She was caged with male pangolin WM6 from 6-10 August. During the caging period, she had mated four times, and given birth on February 12, 2019 with a gestation period 186-189 days. Therefore, she mated and conceived just 11 days after calving and 8 days after the death of her pup. Another example was the female pangolin FG16 gave birth on August 26, 2019, and her pup died on 29th. She then mated with the male pangolin WM9 in cage four times from 3-5 September. She then gave birth on March 8, 2020 with a gestation period of 185-187 days. Therefore, FG16 had mated and conceived eight days after calving and five days after the death of her pup. The reason and mechanism for successful conception of young females in such short period of time after giving birth to and the death of pups is unknown. It may be a reproductive strategy to maintain and expand the population of this species.” with “Pangolin FG6 gave birth on 15 February 2018 and her pup died on 19 March 2018. She mated three times between 9–11 April 2018 and gave birth to a second pup on 6 October 2018. This

indicates that she conceived again 53 days after giving birth and 22 days after the death of her pup. Female FG10 gave birth on July 26, 2018, and her pup died three days later. She was caged with male WM6 from 6–10 August, during which time she mated four times and gave birth on February 12, 2019 after a gestation period 186–189 days. Therefore, she mated and conceived just 11 days after pupping and eight days after the death of her pup. Another example is female FG16, which gave birth on August 26, 2019 and her pup died three days later. She was then placed with a male and mated four times between 3–5 September 2019. She gave birth on March 8, 2020 after a gestation period of 185–187 days. Therefore, FG16 had mated and conceived eight days after pupping and five days after the death of her pup. The reason and mechanism for successful conception of young female pangolins in such a short period of time after giving birth to and the death of pups is unknown. It may be a reproductive strategy to maintain and expand the population of this species.”

Author response: We have modified this part in the revised manuscript. (Lines 380–398)

82. “The proportion of males with mating willingness is low”, Lines 2–3: Change “Of these pangolins, there was three of them (WM6, WM8 and WM9) showed strong willingness for mating” to “Of these pangolins, three (WM6, WM8 and WM9) showed a strong willingness to mate”.

Author response: We have revised this part in the revised manuscript. (Line 405–411)

83. “The proportion of males with mating willingness is low”, Line 4: Change “The male pangolin WM11” to “Male pangolin WM11”.

Author response: We have deleted this part in the revised manuscript. (Line 407)

84. “The proportion of males with mating willingness is low”, Lines 5–7: Change “Another male pangolin WM12 mated with the female pangolin WF2 in April 2017, but did not show any mating after mating with the WF2 pangolin” to “Another male pangolin (WM12) mated with female WF2 in April 2017, but did not show any further mating activity”.

Another male pangolin (WM12) mated with female WF2 several times during the

cohabitation period in April 2017, but since that time, he had not shown any willingness to mate.

Author response: We have deleted this part in the revised manuscript. (Lines 408–410)

85. “Captive pangolins reproduce more female than male offspring”: Change this heading to “Captive pangolins produce more female than male offspring”.

Author response: We have replaced this heading with “Sex ratio of the captive-born offspring” in the revised manuscript. (Line 429)

86. “Captive pangolins reproduce more female than male offspring”, Line 1: Change “sexual ratio” to “sex ratio”.

Author response: We have modified accordingly in the revised manuscript. (Line 428)

87. “Captive pangolins reproduce more female than male offspring”, Line 2: Change “in our base” to “in our facility”.

Author response: We have modified accordingly in the revised manuscript. (Line 428)

88. “Captive pangolins reproduce more female than male offspring”, Lines 3-4: Change “conceived in wild” to “conceived in the wild”.

Author response: We have revised this part in the revised manuscript. (Line 429–430)

89. “Captive pangolins reproduce more female than male offspring”, Line 4: Change “with two offspring were twin” to “with two offspring being twins”.

Author response: We have deleted this part in the revised manuscript. (Lines 430–431)

90. “Captive pangolins reproduce more female than male offspring”, Lines 5-7: Change “Of these 44 offspring, the sex of 38 offspring were identified in time and the remaining failed to identify due to reasons such as offspring died in mothers’ body without identifying their sexes” to “Of these 44 offspring, the sex of 38 were determined while the sex of the remaining individuals could not be determined as they died in vivo prior to being fully developed.”

Author response: Not all deaths were due to this reason, therefore we have slightly revised the proposed sentence. (Lines 431–434)

91. “Survival of wild pangolins and captive-born pangolin offspring”, Line 2: Change “rescue base” to “rescue centre”.

Author response: We have modified accordingly in the revised manuscript. (Line 451)

92. “Survival of wild pangolins and captive-born pangolin offspring”, Line 2: Delete “With our methods”.

Author response: We have deleted it as suggested. (Line 451)

93. “Survival of wild pangolins and captive-born pangolin offspring”, Line 7: Change “base” to “centre”.

Author response: We have modified accordingly in the revised manuscript. (Line 451)

94. “Survival of wild pangolins and captive-born pangolin offspring”, Line 7: Change “survive” to “survived”.

Author response: We have deleted this part in the revised manuscript. (Line 457)

95. “Survival of wild pangolins and captive-born pangolin offspring”, Lines 8-10: Remove these specific references to how long individuals survived – these data are available in your supplementary material. Rather include a summarized version of these data as you have done for captive-bred pangolins (i.e., X individuals survived >500 days, Y individuals survived >1000 days, etc.).

Author response: We have revised this part in the revised manuscript. (Lines 455–460)

96. Figure 3, legend: Delete the sentence “High survival rates of wild pangolins and captive-born pangolins support the view that our methods are considerably good for the rescue and breeding of this mammalian species.” This is an interpretation/opinion based on your graph, and therefore should not form a part of the figure legend.

Author response: We have removed it in the revised manuscript. (Lines 465–467)

97. “Survival of wild pangolins and captive-born pangolin offspring”, Lines 22-24: Replace “Notably, we had first time bred 49 pangolins spanning three filial generations within the 5-year period. With our methods, twenty of them survive up to date (Supplementary Table 2).” with “Notably, during this 5-year period we had successfully bred 49 pangolins spanning three filial generations, 20 of which were still alive at the end of this study (Supplementary Table 2).”

Author response: We have replaced it with “Altogether, we had successfully bred 49 pangolins spanning three filial generations in the five years, 20 of which were still alive at the end of this study” in the revised manuscript. (Lines 477–479)

98. “Survival of wild pangolins and captive-born pangolin offspring”, Lines 27-29: Again, remove these specific references to the number of days that each individual survived – these data are available in your Supplementary Material Table 2. Rather just summarise it as “four individuals survived >900 days, while six individuals survived >1000 days”.

Author response: We have revised this part in the revised manuscript. (Lines 455–456)

99. “Survival of wild pangolins and captive-born pangolin offspring”, Lines 30-31: You cannot make this statement, as you have not provided any information on how you “rescued” these individuals. Rescue goes far beyond maintenance, and includes all veterinary procedures and examinations, medication, etc. You can only say that you were able to maintain both wild-sourced and captive-bred individuals in captivity.

Author response: We have modified it in the revised manuscript. (Lines 485–486)

100. “Captive-born pangolins have better survival rates after weaning”, Lines 5-6: “Notably, the 150 days of pre-weaning period was based on our latest research on captive-born pangolins manuscript in preparation).” You cannot say this, as this is an orphan sentence. Also, there is published literature that states that the weaning period lasts for 90-120 days (Chong et al. 2020. Sunda Pangolin species account, Chapter 6. In in Challender, D.W.,

Nash, H.C. and Waterman, C., 2019. Pangolins: Science, Society and Conservation. Academic Press). If you want to say this you first need to acknowledge the published literature, and then say that your own observations suggest that the weaning period in captivity is longer at 150 days.

Author response: We have modified accordingly in the revised manuscript. (Line 499)

101. Figure 4 caption: Change the caption to read: “Figure 4. Mortality of captive-born pangolins. (a) Mortality in captive-born offspring as a function of ages. (b) Proportion of surviving offspring for two different groups (pre-weaning and post-weaning) across different years.

Author response: We have modified it in the revised manuscript. (Lines 513–518)

102. “Mortality and cause of death”, Line 4. You cannot cite Clark (2008) here. You are reporting on your findings, and as Clark (2008) did not perform the autopsies and subsequently publish the results, you cannot cite her here. Instead, you should cite her in your Discussion when you mention that pyloric ulcers and bronchial complications are commonly seen in pangolin post-mortems.

Author response: This part has been deleted in the revised manuscript.

103. “Mortality and cause of death”, Lines 4-6. “Bacterial isolation of lung and liver tissues showed that most of the detected bacteria were known conditioned pathogens such as Morgellons, Escherichia coli, Klebsiella pneumoniae, and Staphylococcus aureus”. This is the first time that you mentioned molecular characterization of bacteria isolated from lung tissue. Were these tissue samples collected during the post-mortems, and if so, how? Who did the molecular characterization and where? What primers did you use to amplify the pathogen DNA? What were the amplification conditions? How did you determine the identity of the isolated pathogens (e.g., via BLAST Search or phylogenetic inference)? All of this needs to be included in your Materials and Methods section.

Author response: Thank you for highlighting this to us. The method was previously described in our published paper. But we found that it is in Chinese. Therefore, we have

added the method information (English) in the revised Supplementary file to help readers to understand it.

104. “Mortality and cause of death”, Line 1. How many are “some adult individuals”? You need to provide the specific number.

Author response: Thank you for pointing this out. There were 11 adult pangolins. We have modified accordingly in the revised manuscript. (Line 553)

5. “Mortality and cause of death”, Line 7. Change “were dissected” to “was dissected”.

Author response: This part has been deleted in the revised manuscript. (Line 530)

106. “Mortality and cause of death”, Line 16. Change “the presumably” to “the presumed”.

Author response: This part has been deleted in the revised manuscript. (Line 535)

107. “Mortality and cause of death”, Line 17. Change “suspected” to “suspect”.

Author response: This part has been deleted in the revised manuscript. (Line 536)

108. “Mortality and cause of death”, Lines 18-19. Change “pangolins were infected” to “pangolins to be infected by”.

Author response: This part has been deleted in the revised manuscript. (Line 537)

109. “Mortality and cause of death”, Line 19: Please provide the full reference for “Choo, Genome Research”, or else clarify what you mean by this. Is this unpublished data?

Author response: This is our published data. We have moved this part to Discussion section and added the reference in the revised manuscript. (Line 617)

110. “Mortality and cause of death”, Line 20: Change “to make sure the food” to “to ensure that the food”.

Author response: We have revised it in the latest manuscript. (Line 619–620)

111. "Mortality and cause of death", Lines 22-23: Change "For instance, we increased the number of times to clean and wash cages (e.g. 1-2 times/month)" to "For instance, we increased the frequency with which cages were cleaned to 1–2 times per month".

Author response: We have modified it in the revised manuscript. (Lines 620–621)

112. "Mortality and cause of death", Line 23: Change "torches; moreover..." to "torches. Moreover...".

Author response: It has been deleted in the revised manuscript. (Line 656)

113. "Mortality and cause of death", Line 23: What do you mean by "moreover, we cleaned nests immediately when pangolins stop eating"? Does this mean that you cleaned the cages as soon as an individual stopped eating, or was its food removed when it stopped feeding?

Author response: Should be "refuse to eat". We have revised it in the latest manuscript. (Line 623)

114. "Duration of gestation in pangolins": change this heading to "Duration of gestation in Malayan pangolins".

Author response: We have modified accordingly in the revised manuscript. (Line 565)

115. "Duration of gestation in pangolins", Lines 2-4: Replace "When we put the female and male pangolins were placed in a cage for a period, they might mate several times and difficult to determine which mating was successful and lead to conception" with "When females and males were placed together in a cage, they might mate several times and it is therefore difficult to determine which mating was successful and led to conception".

Author response: We have modified accordingly in the revised manuscript. (Lines 567–570)

116. "Duration of gestation in pangolins", Line 8: "with a gestation range of 154-203 days". This differs from the values that you present in Table 2.

Author response: This should be referred to Table 1 (Line 289), instead of Table 2.

117. “Duration of gestation in pangolins”, Lines 8, 15 and 17: How could you determine these gestation periods if in most instances females mated multiple times, ostensibly also while they were pregnant (as reported in your Table 2)? For these values to be accurate, one needs to assume that the first mating always resulted in conception, but I have not seen any evidence to show that this is the case.

Author response: Thank you for pointing this out! Please refer to the response 80 above.

118. Figure 5 caption: Delete “Distribution of the gestation length.”

Author response: We have deleted this part in the revised manuscript. (Line 598)

119. Discussion, Line 2: Change “pangolins” to “Malayan pangolins”.

Author response: We have modified accordingly in the revised manuscript. (Line 602)

120. Discussion, Line 2: Replace “Although previous study reported the breeding” with “Although previous studies have reported the breeding”.

Author response: As suggested, we have modified it in the manuscript. (Lines 602–603)

121. Discussion, Line 3: Replace “to first filial generation” with “to the first filial generation”.

Author response: We have modified accordingly in the revised manuscript. (Line 603)

122. Discussion, Line 3: Delete “however”.

Author response: We have deleted it in the revised manuscript. (Line 604)

123. Discussion, Line 4: “which we summarized in Supplementary Table (Zhang et al, 2015.” Do you mean that you have summarised these differences in a Supplementary Table? If so, which Supplementary Table (provide the number/s). And if that is the case, why is the “Zhang et al. 2015” reference included here?

Author response: Sorry for the confusion. This refers to the supplementary Table 5. The reference Zhang et al. (2015) has been removed in the revised manuscript. (Line 605)

124. Discussion, Line 5: Delete the second “in”.

Author response: We have deleted it in the revised manuscript. (Line 606)

125. Discussion, Lines 6 & 7: “since hundreds of years ago”. This is incorrect, as this has only happened over the past ~150 years. Rather say “over the past two centuries”.

Author response: We have modified accordingly in the revised manuscript. (Line 607)

126. Discussion, Lines 7 & 8: “very long period (e.g. up to 7 years)”. 7 years is not a “very long period” for a species that probably lives for 20-30 years in the wild. If you must say this rather say that you have maintained them in captivity for “an extended period of time”.

Author response: We have modified it in the revised manuscript. (Lines 609–610)

127. Discussion, Lines 10 & 11: “These advances have implications for conservation of endangered pangolins.” If you are going to include this statement, you need to provide evidence on how your study has implications for the conservation of endangered pangolins. Breeding a species in captivity does not necessarily confer a conservation implication – it just means that we can breed them in captivity. For a captive breeding program to confer a conservation benefit it needs to consider such factors as genetic diversity, population genetics, diseases, geographical variation in morphology and genetics and adaptability of the captive-bred individuals to wild conditions, amongst various other factors. It is also pointless to be able to breed an endangered species in captivity if there are no safe areas into which these captive-bred individuals could realistically be released.

Author response: We believe these advances could be important steps for the conservation of the endangered pangolins. One of the success stories is the reintroduction of European bison. European bison hunted to extinction in the wild and survived only in captivity in the early 20th century. The captive European bison were

successfully reintroduced to the wild in the 1950s. Therefore, there are many things we could learn from this success story when reintroducing captive pangolins to the wild in future. We have discussed this in the revised manuscript. (Lines 747–751)

128. Discussion, Line 12: Add a comma after “diseases”.

Author response: We have deleted it in the revised manuscript. (Line 635)

129. Discussion, Line 14: Replace “provide” with “provides”.

Author response: We have deleted it in the revised manuscript. (Line 644)

130. Discussion, Line 32: Replace “observations was that the female pangolins could mate and conceived in very short” with “observations was that female pangolins could mate and conceive in a very short”.

Author response: We have modified accordingly in the revised manuscript. (Line 678)

131. Discussion, Line 33: Replace “co-habiting” with “cohabiting”.

Author response: We have modified accordingly in the revised manuscript. (Line 668)

132. Discussion, Lines 33-34: What is meant by “or the random mating in cages has a higher pregnancy rate”? A higher pregnancy rate than what?

Author response: We have removed this statement. (Line 668)

133. Discussion, Line 34: “It is possible that pangolins are solitary animals”. This is not a possibility, it is a fact and is mentioned in nearly every article dealing with pangolins.

Author response: We have modified accordingly in the revised manuscript. (Line 681)

134. Discussion, Lines 36-37: Replace “In order to breed species, the placental mammal might evolve to have efficient breeding strategies.” with “In order to breed, placental mammals have evolved various breeding strategies.”

Author response: As suggested, we have modified it. (Lines 683–684)

135. Discussion, Lines 43-45: "In other rescue projects of Malayan pangolins in China, it has been understood that the proportion of wild male Malayan pangolin willing to mate is also relatively low." You need to provide a reference for this statement. Which other rescue centres have reported a low willingness to mate among male pangolins?

Author response: The information was obtained from the website. However, the website no longer exists. Therefore, we have deleted this statement in the revised manuscript. (Lines 687–689)

136. Discussion, Line 51: The correct scientific name for a Jaguarundi is *Herpailurus yagouaroundi*.

Author response: We have modified it in the revised manuscript. (Lines 703-704)

137. Discussion, Line 52: The correct scientific name for a lion is *Panthera leo*.

Author response: We have modified accordingly in the revised manuscript. (Line 704)

138. Discussion, Line 52: The correct scientific name for a blackbuck is *Antelope cervicapra*.

Author response: We have modified accordingly in the revised manuscript. (Line 705)

139. Discussion, Line 53: The correct scientific name for an amur leopard is *Panthera pardus orientalis*.

Author response: We have modified accordingly in the revised manuscript. (Line 706)

140. Discussion, Line 53: The correct common name for a pudu is a southern pudu.

Author response: We have modified accordingly in the revised manuscript. (Line 706)

141. Discussion, Line 54: "R.Glatston 1997". Why include the author's initial?

Author response: We have corrected the name in the reference. (Line 817)

142. Discussion, Lines 55-57: Change “Faust and Thompson analyzed the sex ratio of 66 captive mammalian species, they only found two species showing female-biased sex ratios (Faust and Thompson 2000).” to “Faust and Thompson (2000) analyzed the sex ratios of 66 captive mammalian species and found only two species that showed female-biased sex ratios.”

Author response: We have revised it as suggested. (Lines 708–710)

143. Discussion, Line 58: The correct scientific name for the pygmy hippo is *Choeropsis liberiensis*.

Author response: We have modified accordingly in the revised manuscript. (Line 711)

144. Discussion, Line 73: Change “are still rooms for improvement” to “is still room for improvement”.

Author response: We have modified accordingly in the revised manuscript. (Line 727)

145. Discussion, Lines 73-75: Replace “First, it would be interested in future to evaluate the genetic structure of these captive pangolins and compared with wild pangolins” with “Firstly, it would be interesting to evaluate the genetic structure of these captive pangolins and compare this with wild pangolins”.

Author response: We have modified accordingly in the revised manuscript. (Lines 727–729)

146. Discussion, Lines 75-76: “although it may be difficult to get the endangered wild pangolins for analysis”. There is a wealth of genetic data for wild pangolins already, so this statement is incorrect and should be removed.

Author response: We have deleted this part in the revised manuscript. (Lines 729–731)

147. Discussion, Line 77: Replace “planning to look for more male pangolins into our base” with “planning to source additional male pangolins for our centre”.

Author response: We have modified accordingly in the revised manuscript. (Line 731–732)

148. Discussion, Line 78: Replace “preweaning” with “pre-weaning”.

Author response: We have modified accordingly in the revised manuscript. (Line 734)

149. Discussion, Lines 79-80: “Fourth, we will train and gradually release captive-bred pangolins back to the wild.”. This is not a good idea unless it follows the IUCN guidelines for reintroductions. As mentioned above, any reintroduction program is pointless unless there are areas that are secure enough for pangolins to be released into, where there isn’t an existing pangolin population. Furthermore, any reintroduction program needs to consider genetic diversity, population genetics, diseases, geographical variation in morphology and genetics and adaptability of the captive-bred individuals to wild conditions, amongst various other factors.

Author response: We fully agree with your opinion that many factors need to be considered and it is also a good idea to follow the IUCN guidelines. However, there might be not an existing pangolin population currently and we would like to be pioneers to try establishing pangolin population. Notably, the reintroduction of European bison is a good success story that we should learn. This species were hunted to extinction in the wild and survived only in captivity in the early 20th century and was successfully reintroduced to the wild in the 1950s. (Lines 747–751)

150. References: Reference no. 3 (Challender et al., 2019). This is not the complete reference and several author’s names have been omitted from this reference. Please correct this.

Author response: We have addressed it in the revised manuscript. (Line 771)

151. References: “R.Glatston”. Why does one of this author’s initials precede his family name?

Author response: We have corrected it. (Line 817)

152. The references require specific attention. In some instances, the authors initials and surname are provided, while in other instances the authors' entire names and surnames are provided. In at least two instances (Challender et al. 2019 and Heinrich et al. 2016) the authors are incorrectly indicated.

Author response: We have addressed it in the revised manuscript. Please refer to the Reference section in the revised manuscript.

Reviewer #3:

I enjoyed reading this article, and while I think it will make a contribution to the literature, there are a number of areas of the manuscript that require major work if it is to be published in a good quality international journal. I expand on these points below.

I appreciate that English may not be the first language of the authors. However, I would encourage them to check the use of English, or ask an English-speaking colleague to do so, so that they hard work that has gone into this research is clearly conveyed to the reader.

Author response: Sorry for the overall English language quality. We have polished the language in the revised manuscript.

What do the authors mean by systematic in terms of their research? How was the breeding systematic?

Author response: In this study, we have developed a systematic protocol for pangolin held in captivity that include facilities layout, controlled temperature, socialization, feeding, cleaning and so on.

The paper is largely descriptive and would benefit further linking of the results presented to existing knowledge of pangolin husbandry and care, and life history. E.g., why might there be a low willingness to mate among males?

Author response: Thank you for your suggestions! The exact reasons are still unclear, and at present, we have not seen any relevant information published. For example, our

research shows that male pangolins (including wild- and captive-born) in our centre had a relatively low proportion of willingness to mate (wild-born 3/11, captive-born 1/6). However, four males (M6, M8, M9, and SG7) always showed positive willingness to mate and reproduced offspring, while the other males did not follow or crawl across the female. The reason for above phenomenon is still unclear. Nevertheless, we believe that the lower willingness to mate may be related to the captive environment, food, captive management, genetic factors, etc., and more research is needed in the future.

However, it has been reported that captive giant pandas also have the problem of a low proportion of males willing to mate. From 1980-2019, there were nearly 600 captive giant pandas worldwide. However, at the same time, only 26 captive male giant pandas were able to produce offspring through natural mating and the reason is unclear. We have discussed this in the revised manuscript. (Lines 694-701)

What were the causes of death of animals that did not survive in this study?

Author response: From 2016 to 2019, several adult pangolins died each year, but not mass deaths. Through the anatomy of 11 dead pangolins, we found most of them were infected by pathogens. Therefore, from December 2019, we adopted measures such as strengthening sanitation, disinfection, and dry preservation of food. The phenomenon of diseases has been greatly reduced, and adult pangolins no longer die. Therefore, the cause of death may be related to the pathogens in the captive environment and food (please refer to the section “Assessment of possible cause of death” at Lines 520–563 in the revised manuscript).

The authors say ‘keeper management’ but this is not very specific -what is actually meant by such terms? Many of the points made are superficial and not explored in very much depth at all which is disappointing. More detailed thinking here is needed and these such points should be elaborated on.

Author response: Thank you for highlighting this to us. In the revised manuscript, we added more thoughts and discussed these observations in the Discussions section

including the possible reasons for the low willingness to mate and the causes of death. Regarding the “keeper management”, we have removed it in the revised manuscript to avoid confusions.

Can the authors clarify what is happening with weaning?

Author response: Thank you very much for the good question! It has been reported that the weaning period of Malayan pangolin is 3–4 months. However, most pangolin pups in our centre successfully weaned in 4–5 months although we believe that it could be related to the artificial food or environmental factors. The exact reasons are still under investigation.

Fig 4b suggests that once an animal is weaned it survives. But there has been a decline in weaning success between 2016 and 2020. Why is this?

Author response: We have thought about this issue before. We believe that the decline in weaning success between 2016 and 2020 was mainly due to two main reasons. Firstly, the high success rate at the beginning of this study (e.g. 2016) might not be representative due to the small number of captive-born pangolins. Secondly, the decline of weaning success in the middle of the period might be likely due to the severe infections that cause many pangolins died. However, the weaning success had significantly been increased after we implemented more strict measures including keeping the food in a dry and hygienic environment, cleaning the cages immediately when pangolins refuse to eat for a day, etc. We have discussed this in our manuscript.

Minor points:

Title - check the syntax.

Author response: Addressed.

L19/50 – change to “threatened”; same for introduction.

Author response: Addressed. (Line 31)

L22 mating and reproduction?

Author response: This part has been deleted in the revised manuscript. (Line 36)

L28. How are you defining conception rate? Please make this clear for the reader.

Author response: Conception is important to allow reproduction to occur and high rates of conception to allow effective breeding. According to the medical dictionary (Koester. et.al 1999), conception rate is the percentage of matings that result in conception. We have included more information about the conception rate at Lines 268–277 and Lines 331–337 in the revised manuscript.

L36 – how do this captive rate relate to wild populations?

Author response: It is unclear whether the wild populations also have such ratio since the sex ratio of wild populations have not been reported.

L40/80 – can 5 years (less than the lifespan of *M. javanica*) be considered long term?
Suggest revising.

Author response: We have removed it from the latest manuscript. (Line 59)

L46 – the authors should suggest additional key words.

Author response: We have added more keywords as suggested. (Lines 68-70)

53-55 - Check house rules – should you include scientific names?

Author response: We have added scientific names. (Lines 83-85)

58 – Suggest you refer to the most up to date version of the IUCN Red List, which is 2021.

Author response: We have added to the latest version. (Line 112)

60 – there are better references than NGO report- see here:

<https://www.sciencedirect.com/science/article/pii/B9780128155073000034>

Author response: We have deleted this part in the revised manuscript.

74 – Chinese pangolins are now living into their mid-20s in Taipei Zoo – see here:

<https://www.sciencedirect.com/science/article/pii/B9780128155073000368>

Author response: Thank you very much for sharing this. We have added this information in our revised manuscript. (Lines 129-)

78 - this is a more up to date reference -

<https://www.sciencedirect.com/science/article/pii/B9780128155073000289>.

Author response: We have added it in the revised manuscript. (Line 124)

L80 – what is ‘very successful?’

Author response: We have removed it in the revised manuscript. (Line 107)

L110 – So there was case of twins being born? Please clarify in the text.

Author response: Yes, we have clarified it in the revised manuscript. (Line 283)

L23-124 – was the floor wire mesh?

Author response: Yes, you are right. We revised the manuscript accordingly. (Line 193)

L136-138 – in what quantities were each of these items provided?

Author response: We have added this information in the revised manuscript. (Line 199)

The first cage mating has a high conception rate: please explain how first and second caging worked – how many days were there between the first and second caging?

Author response: Since the female's oestrus is not obvious, the time when it is selected to be cohabitated with the male is random. After the first cohabitation, to increase the conception rate, some females and males will be cohabitated more than once. Except for the time interval between first and second cohabitation, which is <1 month, all others were >1 month apart. Some females no longer accept mating after conception, such as F6.

Please refer to the Methods section for more detail. (Lines 232-250)

Please elaborate on why you think the conception rate is high based on knowledge of the species' ecology and biology.

Author response: Since the female pangolin's oestrus is not obvious, the time when it is selected to be caged with the male is random. Of the 18 female offspring, 14 of them were conceived for the first time when the female completed mating, and the other four were conceived by the second mate. The pregnancy rate of the first caged mating was 78% (14/18) (Lines 339-342). Based on our observations, pangolins likely practice copulation-induced ovulation like other animals such as rabbits. When the female is not conceived, she can accept mating and conceive at any time, which has an efficient reproductive strategy. This could be a strategy to enhance their reproductive rate, especially for solitary animals such as pangolins (Lines 674-684).

Why do you think *M. javanica* copulates while pregnant?

Author response: Thank you very much for the good question. We believe that it could be related to the behaviour habit of the species itself because there are many factors that may not be beneficial to the sustainability of the pangolin population, such as solitary animals, one foetus per pregnancy, long pregnancy period, etc. Moreover, our data showed that female pangolins may practice copulation-induced ovulation. Therefore, the females have the habit of accepting mating including during pregnancy, which can likely improve its reproductive efficiency. (Lines 666-684)

Any evidence of females breeding while they are rearing young? This has been reported in wild Chinese pangolins.

Author response: Yes, we did observe this behaviour.

How did you determine willingness to mate among males? Please provide further details.

Author response: After cohabitation, males who have the desire to mate will usually follow, touch, climb, before mating with the females. If there is no desire to mate, the males do not show these behaviours. We have defined this in the revised manuscript.

Please elaborate on why you think there is a low/no willingness to mate among some males? Are there theoretical and/or practical reasons why this might be the case?

Author response: Please refer to our explanation on “mating willingness” above. Our research shows that male pangolins (including wild- and captive-born) in our centre had a relatively low proportion of willingness to mate (wild-born 3/11, captive-born 1/6). However, four males (M6, M8, M9, and SG7) always showed positive willingness to mate and reproduced offspring, while the other males did not follow or crawl across the female. The reason for above phenomenon is still unclear. Nevertheless, we believe that the lower willingness to mate may be related to the captive environment, food, captive management, genetic factors, etc., and more research is needed in the future.

Notably, it has been reported that captive giant pandas also have the problem of a low proportion of males willing to mate. From 1980-2019, there were nearly 600 captive giant pandas worldwide, but only 26 captive male giant pandas were able to produce offspring through natural mating and most captive adult male giant pandas do not mate naturally, and the reason is unclear. We have added this in our revised manuscript (Lines 694-701).

What were the strict measures to improve survival rates? Please elaborate on any changes to care given.

Author response: We did discuss these measures in our manuscript (line 619–627).

Of the animals that did not survive beyond e.g., 150 days – what was the cause of their death?

Author response: There could be several possibilities to explain this. For example, infectious diseases, environmental factors, nursery, etc. We therefore think pups need a more rigorous hygiene environment and nursery. We discuss these possibilities in the revised manuscript (Line 633–638).

Can the authors elaborate on what are the major problems with weaning? This still seems to be causing problems with breeding and rearing *M. javanica*.

Author response: There could be several possibilities to explain this. We did discuss these possibilities in manuscript (Line 633–638).

On deaths – are the respiratory diseases related to not having enough remove in their cages (which seem small) and which could therefore affect lung function?

Author response: We did some experiments before to examine whether pangolins prefer small or large cages. Pangolins tended to prefer the small cages (perhaps they disliked the large cages). Therefore, it is unlikely that the size of the cages causing their deaths.

What are gas torches? Please clarify for the reader.

Author response: It is a liquefied petroleum gas (LPG) flamethrower, which is a heating or welding tool that used for burning. We have clarified this in our revised manuscript. (Lines 193-195).

Change daughters and sons to males and females.

Author response: We have revised it as suggested. (Line 720)

REVIEWERS' COMMENTS:

Reviewer #1 (Remarks to the Author):

Yan et al. describe the self-sustaining captive pangolin population in this manuscript. I think the authors have done a satisfactory job responding to comments from the previous version and am happy with the information and analyses contained in the current version. Below, I highlight only a few very minor suggestions to the text that I hope will increase clarity.

Lines 60-63: it isn't clear to me how trafficking relates to agricultural expansion-related declines. Could you please clarify this?

Lines 71-73: I'm not sure if I follow this sentence, and I think 'powerful' is the part that is throwing me off. Do you mean that captive breeding is an essential method for preventing pangolin extinction?

Line 75: is 12-19 years the maximum survival time? If so this isn't clear.

Line 103: Do you mean pangolins were confiscated *by* law enforcement?

Reviewer #2 (Remarks to the Author):

This manuscript details breeding of captive-maintained Malayan pangolins (*Manis javanica*) to the third filial generation.

This version is an immense improvement over the previous version of this manuscript, and I commend the authors for this. The language is clear and concise, the manuscript structure was appropriate, and I was able to understand all statements and follow all threads of thought without any difficulty - thank you.

Please refer to my specific comments and suggested edits in the accompanying manuscript, and please pay particular attention to the formatting of references. There are several instances of incomplete references, all referring to chapters in the book "Pangolins: Science, Society and Conservation". Please correct these.

With regards to your supplementary material:

I suggest that you rearrange the order of your supplementary material to make it easier for the reader to follow. I suggest that you place all your Supplementary Tables first (each starting on a new page), then all your Supplementary Figures (each on a new page), and lastly your Supplementary Videos (the three video captions can all be grouped on a single page).

Also in your Supplementary material, for all your table and figure legends, only the "Supplementary Table X" or "Supplementary Figure X" portion of the caption should be in bold – the rest of the caption should be normal font (see the main manuscript, where I have also corrected this).

Change Supplementary Table 1's caption to "List of 33 wild-caught Malayan pangolins (*Manis javanica*) used in this study. Survival and mortality were recorded until 30 November 2020. WF = Wild female, WM = Wild male. Initial mass is the mass when the pangolin entered our rescue center."

In Supplementary Table 1, change the column heading "Initial Weight" to "Initial mass".

Change Supplementary Table 2's caption to "List of 49 captive-born Malayan pangolins (*Manis javanica*) used in this study and their status. Survival and mortality were recorded until 30 November 2020. ♀ = female, ♂ = male, ? = unknown, FG = First-generation offspring, SG =

Second-generation offspring, TG = Third-generation offspring.

Change Supplementary Table 3's caption to "Basic information of male Malayan pangolins (*Manis javanica*) used for mating.

In Supplementary Table 3, change both column headings containing "Weight" to "Mass".

Also in Supplementary Table 3, use an 'en-dash' to indicate a range of values (final column).

In Supplementary Figure 1, second-last caption: Place "383 days" in bold font to match the style of the remaining captions.

In Supplementary Figure 1, last two captions: Change the "&" to "and" to match the style of the remaining captions.

Change the caption of Supplementary Table 4 to "Sexual maturity of captive-born Malayan pangolins (*Manis javanica*). Four captive-born female offspring successfully mated with male pangolins and conceived even before they separated from their mothers.

Delete Supplementary Table 5 – It adds no value to your manuscript. I have already worked the main findings of this table into the concluding paragraph of the main manuscript.

Move your "Supplementary methods" into the main manuscript, as I have done.

Reviewer #3 (Remarks to the Author):

I commend the authors for revising the articles in such a comprehensive manner after major reviews. I think the articles is nearly ready for publication but I did notice a number of minor issues that really ought to be addressed before the article can be published. These are detailed below.

L27 – 'in the last century' would be better.

L30 – critically endangered should be Critically Endangered.

L40 – word 'shortly' needed?

L47 – what is meant by 're-understand' pangolins?

L66 - critically endangered should be Critically Endangered. Same for Endangered. And throughout.

L67-69 – It looks like these figures come from this book chapter, which should be references accordingly: <https://www.sciencedirect.com/science/article/pii/B978012815507300006X>.

L137-138 – is this currently planned?

L144 – animal information or pangolin information? As opposed to person.

L146 – 148 – Check the syntax. This reads more as at what you would do than what you did. Check tense.

L149 – i.e., whether you observed the mating behaviour, correct?

L160-162 – check tense ref what you did vs. what you will do. You need to say the former.

L176 – it would be useful here to state here how many dead pangolins you had across the study duration.

L188 – random with respect to date? Please clarify.

L189-195 – this differs to the detail in the abstract. Please ensure coherency between these results and the abstract.

Table 1 – the Remarks columns refers to offspring only, correct? Assuming so, it would be useful to label this column as such.

Table 1 – I recommend double checking the table so it is clear to the reader which information relates to which animal. Including what each colour means in the table caption would help.

L218-219 – does this mean that females that were already pregnant (but perhaps it was unknown at the time) were cohabited with males? Please clarify.

Fig. 2 does not appear to be referred to in the text. Please refer to this figure in the text.

Table 2 - Table 1 – I recommend double checking the table so it is clear to the reader which

information relates to which animal. Including what each colour means in the table caption would help. And which animal the remarks column refers to.

L235-237 – where these animals cohabited with males while with their mothers? Or were they removed from their mothers for cohabitation? Please clarify so that the reader could repeat this if needed.

L237-238 – what does this mean? You mean these animals were first cohabited at a slightly older age?

L217 – no 'discernible' intention to mate might be better? They may have had an intention but it was not possible to detect it.

L299-304 – Some commentators could critique this article as reporting low mortality rates when only 20/49 (41%) young that were bred survived to November 2020, indicating that 59% did not survive. You might want to consider revising the wording here to reflect this. I also recommend referring to this more explicitly in the discussion and recognising the need for further research on survival rates to increase the age at which captive animals survive. For instance, while some animals (e.g., FG2) has survived for 1673, others (e.g., SG13, TG3) survived less than 20 days.

L347 – 177-192 days, correct?

L374 – I would include a sentence here with some more details of the animal that survived for 7 years because this study is only focused on the period 2016–2020 (i.e., less than 7 years).

L378 – you could include a useful example here. E.g., you could state that knowing that if you have females and males (with the males willing to breed), they typically only need to be cohabited for one week together and there is a high chance that they will have mated and that the females will be pregnant. This is useful information for future breeding programmes and management of those programmes, i.e., when to introduce males and females and for how long.

L399 – can you elaborate on nursery? It might be helpful to say that further research on these factors and issues is required?

L401 – what do you propose in terms of extra care?

L409-411 – please include references for these points.

L409-411 – the evidence indicates Sunda pangolins do not breed seasonally, but that some species (e.g., the Chinese pangolin) does breed seasonally. See the work of Wu Shi Bao and Nick Ching-Min Sun.

L437-441 – this a long sentence and is hard to read. Suggest checking for clarity.

L470-472 – for me a key point to emphasise here is further research into survival is needed. This is a research study run over 4-5 years and of the 49 pangolins bred only 41% remain alive to November 2020. My suggestion here is to emphasise that a key focus of future research should be on further understanding the conditions necessary for survival in captivity, and then successful captive breeding.

L472-475 – This seems somewhat out of place and I am not sure if it belongs in this manuscript. This is because the reintroduction of species back into the wild is complicated and there isn't enough room in your article to fully discuss how this would work, the pros, the cons and etc. My suggestion is to remove this section and simply note that successful captive breeding of Sunda pangolins could be used for introduction purposes, which would benefit from following international standards. This reference might be useful: <https://www.iucn.org/content/guidelines-reintroductions-and-other-conservation-translocations>.

Response to Reviewers

REVIEWERS' COMMENTS:

Reviewer #1 (Remarks to the Author):

Yan et al. describe the self-sustaining captive pangolin population in this manuscript. I think the authors have done a satisfactory job responding to comments from the previous version and am happy with the information and analyses contained in the current version. Below, I highlight only a few very minor suggestions to the text that I hope will increase clarity.

Lines 60-63: it isn't clear to me how trafficking relates to agricultural expansion-related declines. Could you please clarify this?

- We have revised sentence. (Lines 58-59)

Lines 71-73: I'm not sure if I follow this sentence, and I think 'powerful' is the part that is throwing me off. Do you mean that captive breeding is an essential method for preventing pangolin extinction?

- We have replaced it with "useful means" (Line 68). Captive breeding can be an useful method to prevent pangolin extinction. We have discussed a successful case (European bison) in the discussion section of manuscript. Please see response in Reviewer 3 L378

Line 75: is 12-19 years the maximum survival time? If so this isn't clear.

- It is the reported longest survival time of pangolins in captivity.

Line 103: Do you mean pangolins were confiscated *by* law enforcement?

- Yes, they were confiscated by law enforcement.

Reviewer #2 (Remarks to the Author):

This manuscript details breeding of captive-maintained Malayan pangolins (*Manis javanica*) to the third filial generation.

This version is an immense improvement over the previous version of this manuscript, and I commend the authors for this. The language is clear and concise, the manuscript structure was appropriate, and I was able to understand all statements and follow all threads of thought without any difficulty - thank you.

Please refer to my specific comments and suggested edits in the accompanying manuscript, and please pay particular attention to the formatting of references. There are several instances of incomplete references, all referring to chapters in the book "Pangolins: Science, Society and Conservation". Please correct these.

- We have corrected it as suggested.

With regards to your supplementary material:

I suggest that you rearrange the order of your supplementary material to make it easier for the reader to follow. I suggest that you place all your Supplementary Tables first (each starting on a new page), then all your Supplementary Figures (each on a new page), and lastly your Supplementary Videos (the three video captions can all be grouped on a single page).

- We have revised it as suggested.

Also in your Supplementary material, for all your table and figure legends, only the “Supplementary Table X” or “Supplementary Figure X” portion of the caption should be in bold – the rest of the caption should be normal font (see the main manuscript, where I have also corrected this).

- We have revised it.

Change Supplementary Table 1’s caption to “List of 33 wild-caught Malayan pangolins (*Manis javanica*) used in this study. Survival and mortality were recorded until 30 November 2020. WF = Wild female, WM = Wild male. Initial mass is the mass when the pangolin entered our rescue center.”

- We have revised it.

In Supplementary Table 1, change the column heading “Initial Weight” to “Initial mass”.

- We have revised it.

Change Supplementary Table 2’s caption to “List of 49 captive-born Malayan pangolins (*Manis javanica*) used in this study and their status. Survival and mortality were recorded until 30 November 2020. ♀ = female, ♂ = male, ? = unknown, FG = First-generation offspring, SG = Second-generation offspring, TG = Third-generation offspring.

- We have revised it.

Change Supplementary Table 3’s caption to “Basic information of male Malayan pangolins (*Manis javanica*) used for mating.

- We have revised it.

In Supplementary Table 3, change both column headings containing “Weight” to “Mass”.

- We have revised it.

Also in Supplementary Table 3, use an ‘en-dash’ to indicate a range of values (final column).

- We have revised it.

In Supplementary Figure 1, second-last caption: Place “383 days” in bold font to match the style of the remaining captions.

- We have revised it.

In Supplementary Figure 1, last two captions: Change the “&” to “and” to

match the style of the remaining captions.

- We have revised it.

Change the caption of Supplementary Table 4 to “Sexual maturity of captive-born Malayan pangolins (*Manis javanica*). Four captive-born female offspring successfully mated with male pangolins and conceived even before they separated from their mothers.

- We have revised it.

Delete Supplementary Table 5 – It adds no value to your manuscript. I have already worked the main findings of this table into the concluding paragraph of the main manuscript.

- We have removed it from the latest manuscript.

Move your “Supplementary methods” into the main manuscript, as I have done.

- Agreed. We have moved it to the main manuscript. (Lines 171-184)

Other comments (found in the attached manuscript from the reviewer)

(1) Were the babies rolled in a ball or stretched out when these measurements were taken? This makes a big difference to the measurement, so it should be mentioned here.

- We measured the stretched pangolins on a special ruler table (see Fig. 1).

(2) Line 304: So you tested 14 wild males. Three males mated and nine did not. What about the other two males? (3 + 9 = 12).

- Thank you very much for highlighting this mistake. We have corrected it in the latest manuscript. (Lines 259-260)

(3) Line 336: Above the figure you say that successfully kept four pangolins >2000 days, 11 pangolins >1500 days and four pangolins >1000 days?

- The figures above refer to wild pangolins, instead of captive-born pangolins.

(4) Line 479: Is it possible that these unwilling males were young, and may therefore not have reached sexual maturity yet?

- It is unlikely because the male SG7 can mate and conceive at 9 months, but the others are more than a year old and did not show willingness to mate. Besides that, FG2 has been born for 5 years, and also did not show willingness to mate.

(5) This should be done in line with the IUCN Guidelines for Reintroductions, including subjecting them to a comprehensive disease-screening and genetic diversity analysis.

- We have added this statement in the latest manuscript. (Lines 453-455)

Reviewer #3 (Remarks to the Author):

I commend the authors for revising the articles in such a comprehensive manner after major reviews. I think the articles is nearly ready for publication but I did notice a number of minor issues that really ought to be addressed before the article can be published. These are detailed below.

L27 – ‘in the last century’ would be better.

- We have revised it as suggested. (Line 27)

L30 – critically endangered should be Critically Endangered.

- Revised. (Lines 29-30)

L40 – word ‘shortly’ needed?

- We have removed it.(Line 39)

L47 – what is meant by ‘re-understand’ pangolins?

- The word of “understand” was used. (Line 46)

L66 - critically endangered should be Critically Endangered. Same for Endangered. And throughout.

- Thank you very much for highlighting this to us. We have corrected it in the latest manuscript. (Lines 62-63)

L67-69 – It looks like these figures come from this book chapter, which should be references

accordingly: <https://www.sciencedirect.com/science/article/pii/B978012815507300006X>.

- Yes, we cited this reference.

L137-138 – is this currently planned?

- This is an on-going plan.

L144 – animal information or pangolin information? As opposed to person.

- We have revised it. (Line 139)

L146 – 148 – Check the syntax. This reads more as at what you would do than what you did. Check tense.

- Thank you very much for highlighting this to us. We have revised these sentences. (Lines 141-143)

L149 – i.e., whether you observed the mating behaviour, correct?

- We have corrected it. (Line 144)

L160-162 – check tense ref what you did vs. what you will do. You need to say the former.

- We have revised it. (Line 156)

L176 – it would be useful here to state here how many dead pangolins you had across the study duration.

- As suggested, we have included this information. (Lines 170-171)

L188 – random with respect to date? Please clarify.

- Yes, please see the method section.

L189-195 – this differs to the detail in the abstract. Please ensure coherency between these results and the abstract.

- The rate of mating and conception rate in abstract referred to captive-born pangolins, whereas L189-195 discussed the rate of mating and conception for wild pangolins in our base.

Table 1 – the Remarks columns refers to offspring only, correct? Assuming so, it would be useful to label this column as such.

- Yes, we have labeled it in the latest manuscript.

Table 1 – I recommend double checking the table so it is clear to the reader which information relates to which animal. Including what each colour means in the table caption would help.

- To avoid confusion, we have decided to remove these colors.

L218-219 – does this mean that females that were already pregnant (but perhaps it was unknown at the time) were cohabited with males? Please clarify.

- Yes, you are right.

Fig. 2 does not appear to be referred to in the text. Please refer to this figure in the text.

- We have changed Fig. 2 to Fig 1 and referred it in the text. (Line 149)

Table 2 - Table 1 – I recommend double checking the table so it is clear to the reader which information relates to which animal. Including what each colour means in the table caption would help. And which animal the remarks column refers to.

- To avoid confusion, we have decided to remove these colors.

L235-237 – where these animals cohabited with males while with their mothers? Or were they removed from their mothers for cohabitation? Please clarify so that the reader could repeat this if needed.

- They were not separated from mothers.

L237-238 – what does this mean? You mean these animals were first cohabited at a slightly older age?

- Yes

L217 – no ‘discernible’ intention to mate might be better? They may have had an intention but it was not possible to detect it.

- I think it should refer to L271. As suggested, we have added it. (Line 261)

L299-304 – Some commentators could critique this article as reporting low mortality rates when only 20/49 (41%) young that were bred survived to November 2020, indicating that 59% did not survive. You might want to

consider revising the wording here to reflect this. I also recommend referring to this more explicitly in the discussion and recognising the need for further research on survival rates to increase the age at which captive animals survive. For instance, while some animals (e.g., FG2) has survived for 1673, others (e.g., SG13, TG3) survived less than 20 days.

- Thank you for highlighting this to us. Perhaps the high mortality rate was mainly contributed by the deaths of pangolins due to infections in 2018. However, the mortality rate has been improved after we implemented strict measure e.g. mortality rate of captive born pangolins in 2020 is 23.1%. However, if we combine the captive born pangolins and wild pangolins, the mortality rate is 14.3%, which is considerably low.

L347 – 177-192 days, correct?

- Thank you very much for highlighting the typo error. We have corrected it. (Line 329)

L374 – I would include a sentence here with some more details of the animal that survived for 7 years because this study is only focused on the period 2016–2020 (i.e., less than 7 years).

-The animal that survived for 7 years is the wild pangolin, instead of captive born pangolins. Note that our captive breeding program started in 2016. But we brought in wild pangolins into our base even before we started the captive breeding program.

L378 – you could include a useful example here. E.g., you could state that knowing that if you have females and males (with the males willing to breed), they typically only need to be cohabited for one week together and there is a high chance that they will have mated and that the females will be pregnant. This is useful information for future breeding programmes and management of those programmes, i.e., when to introduce males and females and for how long.

- We did give example to show how these advances (e.g., captive breeding of pangolins) are important for the conservation of pangolins by using the European bison as an example in another paragraph of the discussion section (Lines 455-460). Due to the word limit, we could not discuss too many examples, therefore we think that it would be better not to include more examples in the discussion. However, we appreciate the good suggestion.

L399 – can you elaborate on nursery? It might be helpful to say that further research on these factors and issues is required?

- Agreed. We have included this useful statement in our latest manuscript. For nursery, we did not have too much manual intervention in this study since this is our first captive breeding program. We believe that the survival rate of offsprings at preweaning could be improved once we have more intervention (e.g., manually feed infants with milk when necessary) and strengthen our nursery management.

L401 – what do you propose in terms of extra care?

- For example, keeping a cleaner environment and strengthen current management of nursery. We plan to have more manual interventions including manually feed babies with milk if their mothers lack breast milk.

L409-411 – please include references for these points.

- These results actually refer to our observations in this study. We have corrected the sentence. (Line 390)

L409-411 – the evidence indicates Sunda pangolins do not breed seasonally, but that some species (e.g., the Chinese pangolin) does breed seasonally. See the work of Wu Shi Bao and Nick Ching-Min Sun.

- Yes, they have reported that Chinese pangolins breed seasonally in the wild. Consistent with our observations, Wu Shi Bao (2015) has also reported that the breeding of Malayan pangolins in captivity is aseasonal. We have revised the sentence to make it more specific to captive Malayan pangolins in the latest manuscript. (Line 390)

L437-441 – this a long sentence and is hard to read. Suggest checking for clarity.

- We have revised it in the latest manuscript. (Line 418)

L470-472 – for me a key point to emphasise here is further research into survival is needed. This is a research study run over 4-5 years and of the 49 pangolins bred only 41% remain alive to November 2020. My suggestion here is to emphasise that a key focus of future research should be on further understanding the conditions necessary for survival in captivity, and then successful captive breeding.

- Agreed. We have added this suggestion. (Lines 446-448)

L472-475 – This seems somewhat out of place and I am not sure if it belongs in this manuscript. This is because the reintroduction of species back into the wild is complicated and there isn't enough room in your article to fully discuss how this would work, the pros, the cons and etc. My suggestion is to remove this section and simply note that successful captive breeding of Sunda pangolins could be used for introduction purposes, which would benefit from following international standards. This reference might be

useful: <https://www.iucn.org/content/guidelines-reintroductions-and-other-conservation-translocations>.

- We agreed that the reintroduction of species back into the wild is complicated. Although it is complicated, this is our ongoing plan and goal to save pangolins. We will definitely attempt to do it based on the guidelines from the IUCN. Therefore, we think it is important to include it in the discussion section.